# Seedless: on-the-fly pulse calculation for NMR experiments

Charles J. Buchanan[1,2], Gaurav Bhole[3,4], Gogulan Karunanithy [2,7], Virginia Casablancas-Antràs [1,2], Adeline W. J. Poh[5,6], Benjamin G. Davis [5,6], Jonathan A. Jones [3] ✉ & Andrew J. Baldwin [1,2,5] ✉

NMR experiments require sequences of radio frequency (RF) pulses to manipulate nuclear spins. Signal is lost due to non-uniform excitation of nuclear spins resonating at different energies (chemical shifts) and inhomogeneity in the RF unavoidably generated by hardware over the sample volume. To overcome this, we present Seedless, a tool to calculate NMR pulses that compensate for these effects to enhance control of magnetisation and boost signal. As calculations take only a few seconds using an optimised GRadient Ascent Pulse Engineering (GRAPE) implementation, this now allows pulses to be generated in a few seconds, allowing them to be optimised for individual samples and spectrometers ("on-the-fly"). Each calculated pulse requires bands of chemical shift to be identified, over which one of 4 transforms will be performed, selected from a set that covers all commonly used applications. Using imaging experiments, we demonstrate our pulses effectively both increase the size of the coil volume and signal-to-noise in all experiments. We illustrate the approach by showing sensitivity gains in 1, 2 and 3D applications suitable for chemical and biological NMR. Seedless provides a means to enhance sensitivity in all pulse sequences in a manner that can be tailored to different samples and hardware being used.

Nuclear magnetic resonance spectroscopy (NMR) is one of the most widely used techniques for atomic resolution characterisation of molecules and biomolecules. NMR experiments, termed pulse sequences, are currently selected from vast libraries, each providing different molecular characterisations. These are (predominantly) sequences of radio-frequency (RF) "pulses" and delays that together conduct specific manipulations of the nuclear spins in the sample. Pulses are themselves constructed from a series of concatenated "rectangular" elements, each of which has a defined "phase" (angle in the xy plane), a central frequency (transition energy), amplitude

(intensity), and duration. The simplest "rectangular" pulses have constant phase, frequency and amplitude.

Maximum sensitivity in experiments requires pulses that have a uniform performance both spatially (all positions in the sample experience the same excitation), and spectroscopically (where the effects of the pulse are identical over some range of chemical shifts). In practical situations, neither of these conditions are truly met. Spatially, a rectangular pulse will have a variation in amplitude of approximately ±5% variation over the sample from the desired value, with the specific values varying with the hardware[1]. Spectroscopically, a rectangular

[1]Kavli Institute of Nanoscience Discovery, Dorothy Crowfoot Hodgkin Building, Oxford, UK. [2]Physical and Theoretical Chemistry, Oxford University, Oxford, UK. [3]Centre for Quantum Computation, Clarendon Laboratory, University of Oxford, Oxford, UK. [4]Department of Chemistry, Princeton University, Princeton, NJ, USA. [5]Rosalind Franklin Institute, Harwell Science and Innovation Campus, Oxford, UK. [6]Department of Pharmacology, Oxford University, Oxford, UK. [7]Present address: Department of Structural and Molecular Biology, Division of Biosciences, University College London, London, UK. ✉e-mail: jonathan.jones@physics.ox.ac.uk; andrew.baldwin@chem.ox.ac.uk

pulse will have an excitation profile described by a sinc function where "perfect" excitation occurs only for nuclear spins whose transition energy is close to the central frequency. As spectrometers are constructed with higher fields, when examining samples that contain a wide range of chemical shifts, and when performing experiments with larger numbers of pulses, these problems become more acute and sensitivity is lost.

To address this, many "shaped" pulses have been developed[2–7], comprising concatenated rectangular elements each of which can have varying amplitude and phase, which can be collectively optimised to produce specific actions such as inversion, excitation or refocusing[2–7] or for wide broadband excitation[8]. The present paradigm remains the same as first introduced for the BURP pulses in 1991[6], which is to calculate an optimised "shape", with desirable characteristics, where total duration/amplitude can be rescaled to fit specific applications at the point of application. The field advanced significantly with the development of GRadient Ascent Pulse Engineering (GRAPE) methods, which allow efficient calculation of the derivatives required for optimisation[9]. Implementations of the GRAPE algorithm typically require proprietary software such as Spinach[10] that requires MATLAB, or are embedded in general frameworks not specifically optimised for performance, such as QuTIP[11] and SIMPSON[12,13]. These tools have created a range of pulses which are freely available[14–23]. More recently, AI algorithms have also been trained for this purpose[24,25]. However, the calculation of each pulse can still take many hours, preventing their properties being matched to specific targeted hardware/samples, and so experiments are constructed from a limited library of underlying pulses. Because altering the sample buffer can drastically alter how fields affect the sample, and because pre-existing pulses can't always perform the exact function needed in an experiment, there isn't always a pulse available for the task at hand.

To move beyond this paradigm, it would be desirable to calculate bespoke pulses on-the-fly when experiments are initiated at an NMR spectrometer, allowing requirements to be matched to a specific sample/experiment/spectrometer. To address this, we have developed Seedless (GRAPE without the seeds), highly efficient open-source software written in C++ that can be easily compiled in Windows, Linux and Mac operating systems. Using Seedless, the requirements for each pulse, including specifications for chemical shift ranges to perform specific rotations, can in principle be directly stored in a pulse sequence, and all required pulses can be calculated in a specific manner within seconds when the experiment is started. We either match or exceed the expected performance of existing GRAPE pulses (Supplementary Note 5), and can generate substantial sensitivity gains in commonly used experiments.

The performance of the Seedless algorithm relies on an efficient implementation of the GRAPE algorithm for isolated spin ½ nuclei that dramatically reduces the number of calculations that need to be performed when optimising pulses (derivations in Supplementary Note 2)[26,27]. In practical applications, a nucleus, amplitude (peak $B_1$ field), duration, carrier frequency (in ppm) and number of segments are specified, together with one or more ppm ranges (bands) each aiming to performing specific transforms (Table 1, usage instructions Supplementary Note 4). We establish using NMR imaging experiments that significant intensity gains can be obtained by compensating for the inherent inhomogeneity present in a sample, and that the effect of this is to increase the effective coil volume of the spectrometer, increasing the signal to noise (Fig. 1A).

We demonstrate the effectiveness of Seedless by adapting several commonly used pulse sequences. By generating a high bandwidth (300 ppm) pulse, we created a method for quantitative 1D $^{19}F$ spectroscopy (Fig. 1B). Using the "suppress" restraint (Table 1), we created a "perfect echo"[28] 1D pulse sequence that gives a protein 1D NMR spectrum on dilute samples with water artefacts reduced to ca. 7.5 μM

levels, a reduction factor $> 10^7$ (Fig. 2A). We generated a $^{15}N$ HSQC spectrum with peak intensities enhanced by a factor of 58% on spectrometer operating at 950 MHz $^1H$ Larmor frequency. And finally, we calculated a series of pulses suitable for triple resonance biological NMR applications, where in $^{13}C$, we exploit independent control of CO, Cα and Cβ groups to generate triple resonance pulse sequences (HNCACO, HNCO, HNCA, HNCOCA) that boost signal to noise. We discuss the strategies needed to implement the pulses, noting that because seedless pulses do not require adjustment of delays to compensate for imperfections, their implementation is very straightforward, yielding spectra with perfectly phased indirect dimensions that do not require baseline correction. Overall, the 8 pulse sequences we optimise here require 54 bespoke Seedless pulses (Supplementary Note 3), all of which are freely downloadable. All are calculated within a few seconds on a 2021 MacBook Pro with a 10 core M1 Pro processor and 16 GB RAM (Supplementary Fig. 4) and detailed usage instructions are provided ("Methods", Supplementary Note 4).

Seedless allows for pulses to be routinely recalculated with adjusted bandwidths at run-time so that experiments can be optimised for specific samples and spectrometers to boost sensitivity in all pulse sequences, entirely removing the need to store pre-designed "shapes" for re-use. Seedless is free for academic use and can be downloaded from https://seedless.chem.ox.ac.uk.

## Results

### Seedless design and implementation

Each pulse in a sequence needs to perform specific manipulations of spins that span specified ppm ranges. These can be grouped into one of 4 types of transforms (Table 1, derived in detail in Supplementary Note 2). Here, we describe the key results, and provide principles that can guide decisions for how individual pulses should be replaced with Seedless pulses ("Methods"), before demonstrating specific applications (Supplementary Note 3).

All pulses within a sequence aim to take a spin from a starting state (or states) $s$ to a target $f$. If the pulse is imperfect, its action will be to create $f'$. The Uhlmann–Jozsa fidelity $F = \left[ \text{Tr}\left( \sqrt{\sqrt{\rho_{f'}}\rho_f} \right) \right]^2$[29,30] allows the similarity of two density matrices ($\rho$) describing $f$ and $f'$ to be measured[29,30], varying between 0 (dissimilar) to 1 (identical), the latter being achieved only when the pulse exactly creates the target state from the starting state. For systems of spin ½ all states can be treated as quantum mechanically "pure", and the fidelity simplifies to $F = \text{Tr}(\rho_{f'}\rho_f)$[29,31]. (Supplementary Note 2.4). The action of a pulse can be mathematically expressed as a time ordered product of $n$ element propagators, $V = V_n V_{n-1} \ldots V_2 V_1$, where each $V_k$ is a unitary transformation matrix describing the action of element $k$ (Supplementary Note 2.2) each parameterised by an amplitude and a phase, and so $F = \text{Tr}\left( V\rho_s V^{\dagger}\rho_f \right)$. Defining $W = \rho_s V^{\dagger}\rho_f$, then the fidelity can be written as a product of the pulse, $V$, and a restraint function, $W$, such that $F = \text{Tr}(VW)$. We use this to construct $I$, the "infidelity" (Table 1), as a cost function whose value is equal to 0 when the required action has been achieved.

Such a S2S action will perform excitation transformations such as $Z \rightarrow -Y$ (one axis control), but should not also be expected to also perform the other two other cardinal transformations expected from a complete rotation of the Bloch sphere, namely $Y \rightarrow Z$ and $X \rightarrow X$ (3 axis control). We can accomplish 3 axis control via a "universal" rotation by defining $U$ as the "desired" unitary transformation propagator and integrating over all possible starting orientations. In this case, the result is identical, but with $W = \frac{1}{2}U^{\dagger}$ (Table 1 and Supplementary Note 2.4). A universal rotation restraint could also be obtained by applying two S2S restraints to act at a single frequency, but our framework accomplishes this at half the cost.

**Table 1 | The 4 restraint types that can be applied to nuclear spins in a Seedless calculation**

| Type | Axes | W | A | I | B | $\frac{dI}{d\phi_j}$ |
|---|---|---|---|---|---|---|
| Universal/Identity (Supplementary Note 2.4) | 3 | $\frac{1}{2}U^\dagger$ | $VW$ | $1 - \mathrm{Tr}(A)$ | $-A$ | $\mathrm{Re}[\mathrm{Tr}(BC_j)]$ |
| State-to-state (Supplementary Note 2.2) | 1 | $\rho_s V^\dagger \rho_f$ | $VW$ | $1 - \mathrm{Tr}(A)$ | $-2A$ | $\mathrm{Re}[\mathrm{Tr}(BC_j)]$ |
| XYcite (Supplementary Note 2.6) | <1 | $\rho_z V^\dagger \rho_z$ | $VW$ | $[\mathrm{Tr}(A)]^2$ | $4\mathrm{Tr}(A)A$ | $\mathrm{Re}[\mathrm{Tr}(BC_j)]$ |
| Suppression (Supplementary Note 2.7) | 1 | $\rho_z X_k^\dagger \rho_z$ | $X_K W_K$ | $1 - \sum_{k=1}^{n} \mathrm{Tr}(A_k)$ | $-2A_k$ | $\sum_{k=j}^{n} \mathrm{Re}[\mathrm{Tr}(B_k C_j)]$ |

These are shown with the effective number of axes on the Bloch Sphere that are being controlled (detailed derivation: Supplementary Note 2, illustration: Supplementary Fig. 1), up to a maximum of 3 (universal). To control more axes, either a longer total pulse duration is required or a higher maximum amplitude. A common framework is exploited in Seedless, where the overall pulse propagator $V$ and a related quantity for gradients $C_j$ (see text) are first calculated looping over all elements of the pulse, followed by evaluation of $W$, $A$ and $B$, together with the infidelity "cost" function $I$. A final loop over all elements is then required to calculate the derivatives $\frac{dI}{d\phi_j}$ (final column), required for efficient optimisation. A universal rotation requires either an axis and angle to be supplied (e.g., 90°x), or the identity operator and represents an idealised rotation of the Bloch sphere (3 axis control), where the transformation is described by the unitary operator $U$. A state-to-state (S2S) pulse requires starting and finishing matrix states $\rho_s$ and $\rho_f$ to be supplied (as X, Y or Z). This will perform the desired transform (1 axis of control). The XYcite restraint follows a different form, as here we are aiming to select "against" having a Z component by the end of the pulse, rather than, as in all other cases, selecting "for" a state, and so the number of axes over a frequency band is less than 1. The "suppression" restraint effectively repeats a S2S infidelity evaluation for each time point during the pulse and so is a relatively demanding computation, where the acting propagator for the pulse up to element $k$ is $X_k$ (see Supplementary Note 4 for detailed usage instructions).

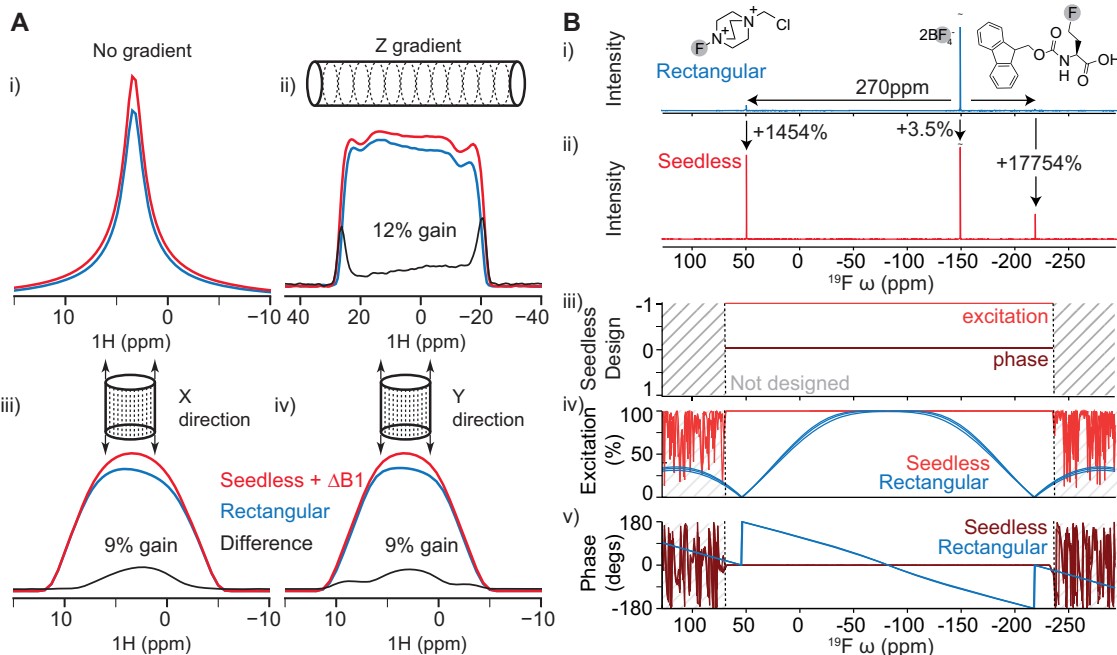

**Fig. 1 | Sensitivity gains from B₁ inhomogeneity compensation, and ultrabroad excitation. A** A ¹³C HSQC imaging gradient echo sequence was designed (Supplementary Note 3.1) and implemented with both Seedless (red) and rectangular (blue) pulses on a sample of ¹³C methanol on a probe equipped with XYZ gradients. **Ai** In the absence of the imaging gradients, the sequence using Seedless pulses delivered substantial gains (12% increase in integral), in a case where all pulses were held precisely on-resonance. **A ii/iii/iv** the Z, X and Y projected images were obtained allowing the enhanced sensitivity to be attributed to regions of space above the top and below the bottom of the sample, at the centre of the tube, precisely where significant divergence is expected in the RF fields generated by the probe. Similar results were obtained for the Z axis when performing the experiment on probes equipped only with Z gradients (Supplementary Fig. 2. **B i** An ultra-broadband

Seedless excitation pulse (Supplementary Note 3.2) was created to analyse sample containing Selectfluor and Fmoc-L-MfeGly, which together contain 3 ¹⁹F environments that span 270 ppm (inset, and "Methods"). Using a rectangular excitation pulse placed in the average location, only the central resonances was appreciably detected (Supplementary Fig. 3). A 2 ms Seedless excitation pulse (Z→ −Y) was designed (**iii**) and the resulting spectrum, (**ii**) contained all three expected resonances perfectly in phase, granting increased intensity of the resonances at the edges of the spectrum by a factor of 1457% and 6400% versus the rectangular pulse. Simulated profiles for the performance of the rectangular and Seedless pulses (**iv, v**) are shown for three different B₁ amplitude values of 0.93, 1.00 and 1.05 of the peak value. Where 3 individual traces cannot be readily discerned the pulse can be considered highly tolerant of amplitude variation.

NMR experiments frequently require different transformations to be applied to a set of frequencies that form contiguous "bands" of chemical shift. Seedless allows a user to specify as many "bands" as required, each with 1 of the 4 types of restraint imposed (Table 1, shown graphically in Supplementary Fig. 1). As a general principle, increasing the number of axes of rotation that a pulse needs to control leads to either an increased total duration or maximum amplitude to retain the same infidelity. Pulses shown in this work are typically operating at the highest possible amplitude allowed unless otherwise

stated (Supplementary Note 3). In brief, the 4 restraints that Seedless can impose are:

1. A "universal" rotation controls of all three rotational 3 cardinal axes of the Bloch sphere, independently of any specific starting and finishing state (Supplementary Note 2.4). These are the rotations anticipated by an idealised on-resonance rectangular pulse, and performed by the central region of an EBURP1[6], REBURP[6] or SURBOP[21] pulses. Pulses of this type are essential when the pulse is expected to handle a range of incoming states such as the refocusing by a 180°

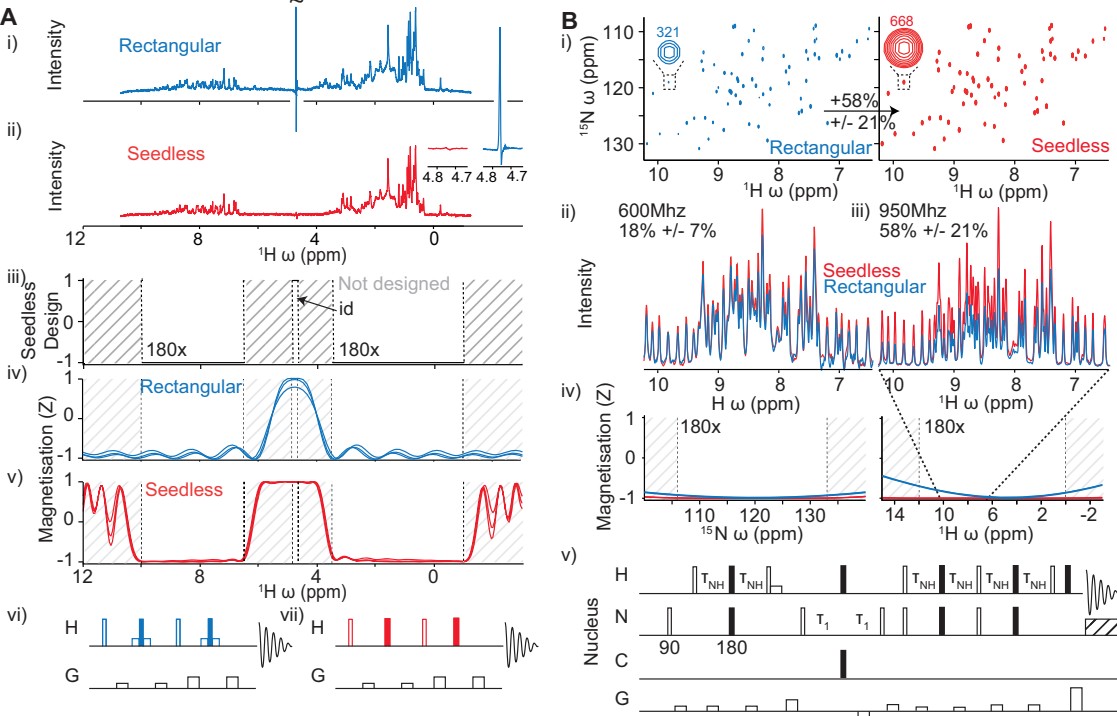

**Fig. 2 | Water suppression and HSQC spectra. A** Watergate perfect echo (PE) pulse sequences using rectangular (**A i, iv, vi**) and Seedless (**A ii, v, vii**) pulses (Supplementary Note 3.3) of 10 µM lysozyme from 64 scans ("Methods"). Signal originating from water (4.7 ppm) when using rectangular pulses has been truncated (tilde). A side-by-side comparison of the complete residual water signal is shown (inset), with S/N of 150 and 2 with rectangular and Seedless pulses, respectively. The ring shifted methyl protons (S/N 4 in both spectra) allowed computation of an effective residual water concentration of 560 µM/7.5 µM (rectangular/Seedless experiments), indicating suppression factors of $10^5$ and $10^7$, respectively. The design requirements for 9.76 kHz Seedless refocusing pulses (unitary rotations on aliphatic and amide bands, suppression of water on the indicated band) are shown (**A iii**), together with simulated performance of a rectangular soft/hard/soft 180° composite pulse and the Seedless pulse (**A iv**, simulations at $B_1$ fields of 0.93, 1.00 and 1.05 are shown, hard pulses at 24.5 kHz, soft at 201 Hz). **B** An application to $^{15}N$ sensitivity enhanced HSQC spectra (**B v**, Supplementary Note 3.4), comparing rectangular (blue) and Seedless (red) pulses acquired on a cryogenically cooled triple resonance probe operating at 950 MHz (**B i, iii**) on a sample of U[$^1H/^{13}C/^{15}N$] ABP1P ("Methods"). Substantial gains are obtained using Seedless pulses, which on average were 58% enhanced for each resonance (**B iii**). A strong correlation is obtained of gains versus $^1H$ chemical shift (Supplementary Fig. 5). Significant gains of 18% were obtained on a room temperature probe at 600 MHz (**B ii**), that show no dependence on $^1H$ chemical shift (Supplementary Fig. 5).

pulse at the centre of a spin-echo, or where magnetisation aligned with two axes need to be simultaneously handled, such as during the refocus INEPT of a sensitivity-enhanced HSQC (Supplementary Note 3.4). An axis and an angle need to be supplied to use this restraint (e.g., $90_x$ or $180_y$, Table 1 and Supplementary Note 2.4). Similarly, an "identity" restraint can be imposed using the same formalism that returns spins in the same orientation as they started. This can be visualised as a pulse that rotates a spin about the *Z*-axis backwards by the exact amount it would have otherwise rotated in the XY plane due to chemical shift evolution during the pulse (Supplementary Note 2.4). This is critical during indirect evolution periods where we seek to decouple the effects of certain nuclei in a manner, while "holding up" evolution of the spin of interest (Supplementary Note 3.4–8) to yield perfectly phased spectra. By removing the need to account for undesirable uncontrolled evolution of chemical shift during pulses, implementations of pulse sequences can be simplified.

2. While the universal rotation is the most versatile restraint, it is also expensive in terms of total time/amplitude required. If the desired transformation can be simplified to known start and finishing states, e.g., $Z \rightarrow -Y$ then a S2S transform can be applied (Table 1 and Supplementary Notes 2.3/4). As this requires only one effective axis of control, excitation ($Z \rightarrow -Y$), de-excitation ($Y \rightarrow Z$), inversion ($Z \rightarrow -Z$) and holding ($Z \rightarrow Z$) pulses will be shorter than their unitary equivalents, the latter two being particularly useful for the passive spin during INEPT transfers, or for decoupling. These are desirable in cases where either relaxation or hardware restraints favour shorter pulses such as the triple resonance sequences described here.

3. If we want to perform a 90° rotation on nuclear spins that either start or finish on Z, but we do not mind what specific phase they end up with in the XY plane, an "XYcite" restraint can be applied. Such a situation arises during INEPT transfers where the duration spent in the XY plane is important but the specific phase is not, which is similar to how the PC9 pulses act[4]. Here, the cost function selects for "not Z" as the final state (Supplementary Note 2.6). Because these are even less restrictive than S2S pulses, these can be shorter, and are used in pairs where the first is a time/phase reversed version of the second[32].

4. Finally, there are cases where we wish to minimally perturb a spin at all stages during a pulse. For such a "suppression", the pulse can be broken into *n* sub-pulses $V_1$, $V_2V_1$, $V_3V_2V_1$ and so on, and the "hold" S2S restraint $Z \rightarrow Z$ is applied for each (Supplementary Note 2.7), similar to the approach of "optimal tracking" designed for heteronuclear spin systems[33]. This is a highly demanding restraint as it takes a fidelity calculation from scaling with *n*, to scaling with $n^2/2$, but is highly suitable for keeping water on the *Z* axis, avoiding effects of radiation damping by preventing transverse magnetisation build up in the XY plane at any point in time. We use this to construct a "perfect echo" 1D pulse sequence with a water suppression factor of $10^7$ (Fig. 2A).

To render the pulses tolerant to amplitude inhomogeneity, we further ensemble average the infidelity over a user specified amplitude distribution. We find that this can be efficiently accomplished on common hardware by sampling the distribution at just three values, at

0.93, 1.00 and 1.05 of the main field strength, weighted $\frac{1}{4}, \frac{1}{2}, \frac{1}{4}$ respectively[1], reflecting the inhomogeneity distributions measured by a nutation experiment. This can always be measured and adjusted to match any system, but works very well here on spectrometers operating with a range of field strengths and with both cryogenically cooled, and room temperature probes ("Methods"). We demonstrate that this substantially raises the S/N by effectively increasing the coil volume of a probe (Fig. 1) in multi-nuclear NMR experiments.

To generate an optimised pulse, both the phase and the amplitude of each of its $n$ rectangular elements could be optimised to lower the infidelity. The response function of a spectrometer however, is not perfect, and when either the phase or amplitude varies suddenly, there are oscillations around the desired value that eventually dampen out to the target value, typically taking around 100 ns[34]. In this work, to partially mitigate against this we focused on constant amplitude "phase-only" pulses, with a minimum duration of each step being 1–2 μs[26,27]. These are desirable as calculation of the gradients becomes particularly efficient (Supplementary Note 2) and fewer steps are required for convergence when compared with hybrid amplitude/ phase optimisation, as has been noted previously[35,36]. Moreover, the resulting pulses tend to emerge from calculations in a "smooth" form removing any need to impose additional restraints to enforce this. In all cases explored here, the inclusion of simultaneous amplitude modulation resulted in pulses with higher infidelity than the phase-only variants. Results in this paper are restricted to constant amplitude "phase-only" pulses, analogous to using frequency modulated (FM) radio transmission rather than amplitude modulated (AM).

The Seedless algorithm also benefits from two further optimisations (Supplementary Note 2.8/9). As we are dealing with calculations for spin ½ nuclei, analytical expressions for the propagators can be used as derived previously[31] (Supplementary Note 2.11). To naïvely evaluate these requires many $2 \times 2$ complex matrix multiplications. Because the matrices are "scaled-unitary" (Supplementary Note 2.12), we can exploit symmetry and halve both the number of multiplications required and the memory storage requirements, transforming all matrix/matrix products into vector/vector multiplications. Similarly, taking the trace of the product of two $2 \times 2$ matrices should naively require 16 multiplications, but we can account for the "scaled-unitary" symmetry above and reduce this to 4 (Supplementary Note 2.12), which particularly accelerates the expensive calculation of the derivatives.

Finally, because all the restraints can be written in the form Tr$(VW)$ (Table 1 and Supplementary Note 2), Seedless uses a common framework requiring only two situations where we first perform a costly loop over all $n$ pulse elements. Here, the partial products at each step, $X_j = V_j X_{j-1}$ are evaluated iteratively, together with a quantity required for the derivatives, $C_j = X_j^{\dagger} \frac{dV_j}{d\phi_j} X_{j-1}$ both of which are retained for each element $j$ (Supplementary Note 2.5). Following this loop, $W$, $A$ and $B$ are calculated together with the infidelity cost function $I$ (Table 1), with the precise form depending on the specific restraint acting on each spin. A final loop over all elements of the pulse calculates the gradients required for optimisation from $\frac{dI}{d\phi_j}$ using $B$ and $C_j$. By taking control of the manipulations at a low level, we achieve highly efficient computation. The program makes use of libBFGS, a C++ implementation of the Broyden-Fletcher-Goldfarb-Shanno algorithm[37–40], which is well suited to this type of optimisation[41]. The code is optionally parallelised using openMP, where each independent frequency and $B_1$ inhomogeneity condition are run on a single core, enabling linear rate enhancements with the number of CPU cores. The program is controlled via simple input scripts (Supplementary Note 4), where lists of pulses for a single experiment can be produced in batch. The program optionally produces reports showing the variation in phase/amplitude of the resulting pulses, together with their performance, which was used to generate the figures in this paper and supplementary information.

The resulting package can calculate band selective $^{13}C$ pulses within a few seconds on a 2021 MacBook Pro with a 10 core M1 Pro processor and 16 GB RAM. To explore the capabilities of "on-the-fly" pulse calculation, we next generated 8 Seedless optimised pulse sequences spanning a range of common applications in both chemical, and biomolecular NMR, and conducted detailed experiments to ascertain the origins of the enhanced sensitivity that was observed.

## Seedless applications

We first considered optimal methods for compensating for the $B_1$ field inhomogeneity inherent in modern probes. This can be easily measured using nutation experiments, and well approximated by a distribution containing 3 fields, at 0.93, 1.00 and 1.05 of the main field, weighted at ¼, ½ and ¼, respectively[1]. Starting with a $^{13}C$ enriched methanol sample ("Methods"), we performed a 1D $^{13}C$ HSQC spectrum with on-resonance rectangular (Fig. 1A, blue), and Seedless pulses (Fig. 1A red) with and without amplitude compensation (+/-$B_1$), on an NMR spectrometer with a $^{1}H$ Larmor frequency of 600 MHz using a room temperature 5 mm HCN probe equipped with XYZ gradients. The data acquired using the Seedless pulses (+ $B_1$) was 12% more intense than data acquired using rectangular pulses (Fig. 1Ai). To visualise specifically which spins were providing additional signal, we adapted a $^{13}C/^{1}H$ HSQC imaging pulse sequence to provide Z, X and Y axis projected images (Supplementary Notes 1, 3.1). Taken together, the projections revealed that Seedless pulses were generating more signal predominantly from spins at the top and bottom of the sample, at the centre, which is where we expect the field lines generated by a saddle coil in the probe to diverge (Fig. 1. Aii). Similar results were seen when testing on probes equipped only with a Z gradient (supplementary Fig. 2). The signal intensity generated from Seedless pulses without $B_1$ compensation were comparable to that achieved from a rectangular pulse revealing that the action of the amplitude compensation is to allow us to detect some parts of the sample that were otherwise being lose, and to effectively increase the effective coil volume, and so provide enhanced signal intensity. We adopted this compensation strategy in all subsequent pulse designs, which worked equally well on all spectrometer and probe combinations tested in this work (600 and 950 MHz spectrometers with room temperature and cryogenic cooled probes, see "Methods").

Seedless allows ultra-wide bandwidth pulses to be calculated. The total range of $^{1}H$ chemical shifts typically experienced in biological and chemical applications spans ca. 10 ppm, which can be easily excited with a 10 μs rectangular pulse, at an amplitude of 25 kHz, easily generated by modern hardware. By contrast, compounds of interest for $^{19}F$ however, can span 300 ppm, which cannot be appreciably excited using rectangular pulses (Fig. 1B iv). We sought to design a Seedless ultra-wide bandwidth pulse suitable for quantitative analysis[8]. We analysed a sample required for a chemical biology application containing Selectfluor (containing two $^{19}F$ species with chemical shifts 47.89 ppm and −151.5 ppm, methods) and a non-natural amino acid analogue Fmoc-L-MfeGly ($^{19}F$ chemical shift −221.5 ppm, synthesis described in "Methods"). Using a rectangular pulse we could only appreciably discern 1 of the 3 species (Fig. 1B blue). We instead designed a 2 ms 25 kHz Seedless S2S (Z → −Y) excitation pulse designed to span a 300 ppm bandwidth (less than 10 s calculation time, Supplementary Fig. 4). The Seedless pulse generated an excellent spectrum, with a similar relative intensity ratio of the three species to that obtained from using 3 separate experiments each on resonance with the individual species (supplementary Fig. 3). The Seedless spectra were perfectly phased, and did not require baseline correction, giving uniform excitation over the expected range of chemical shifts (Fig. 1Biii, iv, v and Supplementary Fig. 3), which was not the case for the spectra acquired using rectangular pulses. Owing to the excitation profile of the rectangular pulse decreasing sharply at the edge of the spectrum, the signal intensity of elements at the edge of the spectrum

was increased by factors exceeding ca. $10^3$ (Fig. 1Bi, ii and Supplementary Fig. 3).

To further test the applicability of Seedless pulses on more complex biological targets, we prepared a sample of 7 kDa U-[$^1$H-$^{15}$N-$^{13}$C] Yeast Actin-Binding Protein 1 (ABP1P, "Methods")[42]. We first compared a sensitivity enhanced $^{15}$N HSQC from the Bruker standard library using rectangular pulses to a Seedless optimised version where all $^1$H/$^{13}$C/$^{15}$N pulses were replaced with Seedless pulses using both a room temperature probe on 600 MHz system and a cryogenically-cooled probe at 950 MHz (detailed description in Supplementary Note 3.4). For ease of implementation, all $^1$H/$^{15}$N pulses were unitary rotations, and the decoupling $^{13}$C 180° pulse was a S2S $Z \rightarrow -Z$ inversion, requiring 5 Seedless pulses in total (Table 2) calculated in under 10 s. Pulses were typically 250 µs in duration, at the maximum permitted amplifier power. The corresponding field was determined, on a sample specific fashion from the 90° pulse times, which for $^1$H was 20/23 kHz, for $^{15}$N was 6.85/7.2 kHz and for $^{13}$C was 17.6/17.6 kHz at 600/950 MHz, respectively.

Individual resonances were compared, and at 600 MHz, an average sensitivity gain of 18% was obtained. The gains were independent of the $^1$H chemical shift (Fig. 2B and Supplementary Fig. 5), indicating that the benefits arise primarily from $B_1$ compensation. By contrast, at 950 MHz, the average gains were more substantial, reaching 58% on average (Fig. 2B), which were linearly correlated with $^1$H chemical shift (Supplementary Fig. 5), indicating that both $B_1$ compensation and improved excitation profiles were together providing sensitivity gains.

In principle, the $^1$H and certain $^{15}$N 90° pulses in the sequence could have been replaced with shorter $Z \rightarrow -Y$ excite/ $Y \rightarrow Z$ de-excite pulses. We note that two 90° pulses in the sensitivity enhanced refocused INEPT period, the first $^{15}$N and the second $^1$H (Supplementary Note 3.4) must perform two simultaneous rotations, either excite/de-excite, and the "hold" $X \rightarrow X$, operation and so these pulses must perform universal rotations.

We next sought to design a novel type of pulse motivated by the challenges associated with water suppression. Much of chemistry and biochemistry occurs in water and so it is frequently desirable to generate spectra that allow us to distinguish molecules of interest, as low as ca. 1 µM, from 55 M water. The "perfect echo" (PE) 1D sequence[43] based on two concatenated "watergate" elements[44], is an excellent way to do this. Using this sequence, a spectrum containing 10 µM hen egg white lysozyme (HEWL) in PBS with 10% $D_2O$ yields a water signal with a S/N of 150 in 64 scans (6 min). By taking a ring-shifted methyl proton from lysozyme (S/N = 4 from 3 protons at 10 µM), we can estimate an effective water concentration of 560 µM (Fig. 2A), indicating a suppression factor of $10^5$.

We then generated Seedless pulses that performed unitary rotations of the aliphatic (−1 to 3.5 ppm) and amide (6.5 to 11 ppm) bands that aimed to leave water minimally perturbed at the end of each of the individual rectangular elements within the pulse (Supplementary Note 3.3) and so avoid effects of radiation damping associated with building up high levels of magnetisation from water in the XY plane for sustained periods. Using this strategy, the S/N of water was reduced to 2 while leaving the signal in the aliphatic and amide regions unchanged, indicating an effective water concentration of 7.5 µM and a Seedless-enabled suppression factor of $10^7$ (Fig. 2A). The "suppression" pulses were relatively demanding calculations (taking ca. 20 s, supplementary Fig. 4). Unlike any other pulse discussed in this manuscript, it was desirable here to use lower amplitudes, ca. 10 kHz to ensure water is minimally perturbed during excitation (Supplementary Note 3.3). The specific trajectories of water reveal relatively brief periods where water does experience brief excitation (Supplementary Note 3.3) but is returned by the pulse to the $Z$ axis to yield excellent protein 1D NMR spectrum at 10 µM with no distortion of the baseline.

Next, we turned to triple resonance pulse sequences, an important family for biomolecular NMR analysis where couplings between $^1$H,$^{15}$N and $^{13}$C nuclei are used to transfer signals along the chains[45,46] to provide residue and atomic resolution information. Sample concentrations of labelled biomolecules are typically lower than those in chemical applications and to accumulate sensitivity, experiment times can extend to days and weeks. We sought to re-design the triple resonance experiments to take advantage of Seedless pulses (Fig. 3) to boost sensitivity. This was achieved via several strategies.

The range of chemical shifts spanned by $^{13}$C in proteins poses both a challenge and an opportunity for triple resonance pulse sequences (as was noted in their original design[46]). Carbonyl (CO) positions resonate in the range 165–185 ppm, and Cα positions fall in the range 40–65 ppm (Fig. 3Ai). The Cβ position has serine and threonine residues resonating in the range 60–90 ppm, with the remainder falling in the range 30–50 ppm. Side chain carbon residues then tend to decrease in chemical shift, reaching ca. 10 ppm for the δ methyl carbons of ILE residues (Fig. 3Ai). High field spectrometers do not allow for uniform excitation from the range 10 ppm to 185 ppm with rectangular pulses (Fig. 3Aii) and triple resonance NMR pulse sequences require the different bands (CO, Cα, Cβ) to be treated independently. Owing to the overlap between the Cα and Cβ chemical shifts, the two cannot be handled perfectly independently in a general case.

Selective excitation was originally accomplished either by using rectangular pulses where the maximum excitation is in the centre of the band of interest and a "null" excitation at the centre of the undesired band ("Methods"), and more recently using selective shaped pulses, such as the Q pulses (e.g., sequences in the Bruker standard library). In both cases, the excitation profiles are not uniform, and spins can evolve during the pulses, causing artefacts and loss of signal intensity (Fig. 3Aiv). To partially compensate for these effects, additional delays and pulses are introduced within sequences, complicating their implementation. To generate a family of $^{13}$C pulses for triple resonance applications using Seedless, three chemical shift bands for CO, Cα and Cβ were defined, where each band is instructed to provide either a universal rotation, a S2S transform or the identity operation (Table 1, specific examples Tables 2 and 3).

We focused on the original four triple resonance experiments, which are the most widely used for backbone resonance assignment, the HNCO, HNCA, HN(CA)CO and HN(CO)CA[46] (for detailed considerations, see Supplementary Notes 3.5–8). These sequences are combinations of INEPT-style transfers and evolution periods, where the pulses either couple or decouple different pairs of spins. Following a process of largely trial and error testing different pulse types in different situations, we established a series of general principles to achieve optimal sensitivity. The speed at which Seedless could generate pulses allowed us to reliably test ca. 50 permutations of each sequence. Using the UnidecNMR peak picking program[47] we could easily compare the intensities from the various 3D spectra to perform this type of screen.

Owing to the relatively high amplitude of $^1$H pulses (23 kHz) and narrow bandwidth, all these pulses could be both relatively short (100 µs) and perform universal rotations. For optimal sensitivity, owing to the lower amplitudes for $^{15}$N and $^{13}$C (6.85 kHz and 17.6 kHz, respectively), pulses should be provided for the specific task, and not be "over specified", and avoiding using longer universal rotations where possible. For phase coherent excitation/de-excitations, S2S $Z \rightarrow -Y / Y \rightarrow Z$ transfers of an acceptable quality were shorter than universal equivalents. For 180° pulses, during INEPT periods and $^{15}$N evolution, the active (transverse) spin always requires a unitary 180° for refocusing, as the pulse needs to correctly transform a range of different incoming phases in the XY plane. Passive spins (held longitudinally) can either have transfer enabled during an INEPT via an S2S inversion ($Z \rightarrow -Z$) or decoupled by applying a "hold" S2S ($Z \rightarrow Z$) transform, avoiding the need to perform the more expensive unitary operations. Exceptions to this minimalist principle are found in the sensitivity enhanced refocused INEPT as described earlier for the

**Table 2 | Summary of the 8 pulse sequences adapted in this work, together with the 54 pulses that were used with their amplitudes (fields) and durations**

| Manuscript Location | Description/Hardware | Specific pulses | Field (kHz) | Duration (µs) |
|---|---|---|---|---|
| Fig. 1A | Imaging (XYZ gradient, RT probe 600 MHz) | $^{1}$H u90x | 31.7 | 160 |
| | | $^{1}$H u180x | 31.7 | 160 |
| | | $^{13}$C u90x | 17.36 | 160 |
| | | $^{13}$C u180x | 17.36 | 160 |
| S3.1 | Imaging (HCN Z gradient RT probe, 600 MHz) | $^{1}$H u90x | 31,7 | 160 |
| | | $^{1}$H u180x | 31.7 | 160 |
| | | $^{13}$C u90x | 17.36 | 160 |
| | | $^{13}$C u180x | 17.36 | 160 |
| Figs. 1B and S3.2 | 19 F 1D (CPRHe-QR cryo-probe, 600 MHz) | $^{19}$F z-y | 20.2 | 2000 |
| Figs. 2A and S3.3 | Perfect echo, water suppressed $^{1}$H 1D (HCN RT probe, 600 MHz) | $^{1}$H u90°x CH/NH+suppress | 9.76 | 4000 |
| | | $^{1}$H u180°x CH/NH+suppress | 9.76 | 4000 |
| Figs. 2B and S3.4 | $^{15}$N sensitivity enhanced HSQC (HCN RT probe, 600 MHz) | $^{1}$H u90x | 23.0 | 100 |
| | | $^{1}$H u90x | 23.0 | 150 |
| | | $^{1}$H u90x | 23.0 | 250 |
| | | $^{1}$H u180x | 23.0 | 100 |
| | | $^{1}$H u180x | 23.0 | 250 |
| | | $^{13}$C CO z-z C$_\alpha$/C$\beta$ z-z | 17.6 | 2000 |
| | | $^{15}$N u90x | 6.85 | 250 |
| | | $^{15}$N u180x | 6.85 | 250 |
| | | $^{15}$N z-y | 6.85 | 250 |
| Figs. 2B and S3.4 | $^{15}$N sensitivity enhanced HSQC (TCI cryo probe, 950 MHz) | $^{1}$H u90x | 17.6 | 250 |
| | | $^{1}$H u180x | 17.6 | 250 |
| | | $^{13}$C CO z-z C$_\alpha$/C$\beta$ z-z | 20.0 | 250 |
| | | $^{15}$N u90x | 7.2 | 250 |
| | | $^{15}$N u180x | 7.2 | 250 |
| Figs. 3B–5 and S3.5 | HNCO (HCN RT probe, 600 MHz) | $^{13}$C CO z-z C$_\alpha$/C$\beta$ zz | 17.6 | 130 |
| | | $^{13}$C CO z-y C$_\alpha$/C$\beta$ zz | 17.6 | 80 |
| | | $^{13}$C CO Id C$_\alpha$/C$\beta$ z-z | 17.6 | 200 |
| | | $^{13}$C CO yz C$_\alpha$/C$\beta$ zz | 17.6 | 80 |
| | | $^{13}$C CO/C$_\alpha$/C$\beta$ z-y | 17.6 | 200 |
| Figs. 4 and S3.6 | HNCA (HCN RT probe, 600 MHz) | $^{13}$C CO zz C$_\alpha$/C$\beta$ z-z | 17.6 | 130 |
| | | $^{13}$C CO zz C$_\alpha$/C$\beta$ z-y | 17.6 | 80 |
| | | $^{13}$C CO/C$\beta$ z-z C$_\alpha$ Id | 17.6 | 400 |
| | | $^{13}$C CO zz C$_\alpha$/C$\beta$ yz | 17.6 | 80 |
| | | $^{13}$C CO z-z C$_\alpha$/C$\beta$ zz | 17.6 | 130 |
| | | $^{13}$C CO/C$_\alpha$/C$\beta$ z-y | 17.6 | 130 |
| Figs. 3B, 4 and S3.7 | HNCACO (HCN RT probe, 600 MHz) | $^{13}$C CO zz C$_\alpha$/C$\beta$ z-z | 17.6 | 130 |
| | | $^{13}$C CO zz C$_\alpha$/C$\beta$ z-y | 17.6 | 80 |
| | | $^{13}$C CO z-z C$_\alpha$ u180x C$\beta$ zz | 17.6 | 400 |
| | | $^{13}$C CO zz C$_\alpha$/C$\beta$ yz | 17.6 | 80 |
| | | $^{13}$C CO z-y C$_\alpha$/C$\beta$ zz | 17.6 | 80 |
| | | $^{13}$C CO Id C$_\alpha$/C$\beta$ z-z | 17.6 | 200 |
| | | $^{13}$C CO yz C$_\alpha$/C$\beta$ zz | 17.6 | 80 |
| | | $^{13}$C CO z-z C$_\alpha$/C$\beta$ zz | 17.6 | 130 |
| | | $^{13}$C CO/C$_\alpha$/C$\beta$ z-y | 17.6 | 200 |
| Figs. 4 and S3.8 | HNCOCA (HCN RT probe, 600 MHz) | $^{13}$C CO z-z C$_\alpha$/C$\beta$ zz | 17.6 | 130 |
| | | $^{13}$C CO z-y C$_\alpha$/C$\beta$ zz | 17.6 | 80 |
| | | $^{13}$C CO u180x C$_\alpha$/C$\beta$ z-z | 17.6 | 200 |
| | | $^{13}$C CO yz C$_\alpha$/C$\beta$ zz | 17.6 | 80 |
| | | $^{13}$C CO zz C$_\alpha$/C$\beta$ z-y | 17.6 | 80 |
| | | $^{13}$C CO/C$\beta$ z-z C$_\alpha$ Id | 17.6 | 400 |
| | | $^{13}$C CO zz C$_\alpha$/C$\beta$ yz | 17.6 | 80 |
| | | $^{13}$C CO zz C$_\alpha$/C$\beta$ z-z | 17.6 | 130 |
| | | $^{13}$C CO/C$_\alpha$/C$\beta$ z-y | 17.6 | 200 |

For the specific pulses, the restraints are summarized. u90x, u180x and Id are "universal" pulse restraints mapping to a 90°x, a 180°x, and the identity operation. Where operators are written with a cardinal axis, the S2S operation is implied, e.g., "z-z", which should be read as "start on z, finish on –z". In all cases, unitary pulses with 3 axis control for each band could have been used, but if the manipulations of spins within a pulse sequence can instead be accomplished with a S2S transform, then the resulting pulse can be made shorter than the unitary equivalent. This is particularly for the triple resonance $^{13}$C pulses. In the location column, S implies "Supplementary Note" for the relevant detail.

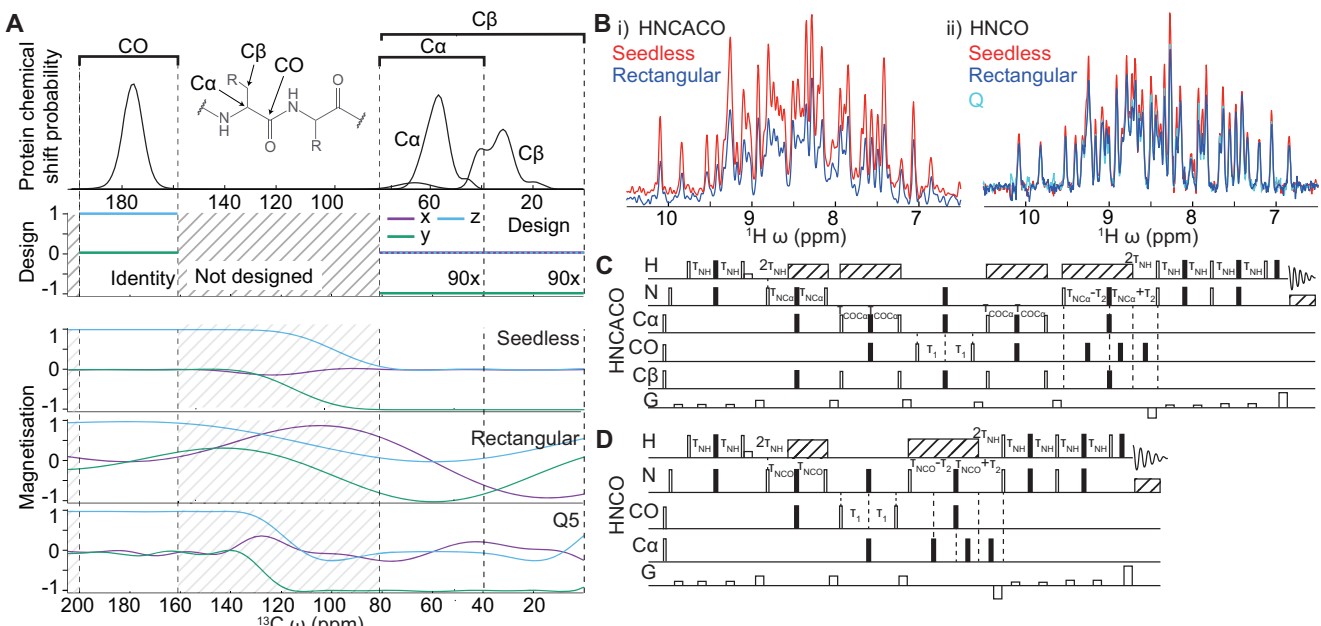

**Fig. 3 | Triple resonance applications. A i** The expected distribution of ¹³C chemical shifts expected in proteins (data from the BMRB[50]). Triple resonance sequences exploit the differences in chemical shift between CO, and Cα/Cβ residues for selective transfers (a schematic amino acid is shown). The overlap of the Cβ and Cα chemical shifts renders it impossible to independently control and hence decouple both groups uniformly for all residues. **A ii** a simulation of a spin initially aligned on the Z axis following action of a Seedless pulse, a rectangular pulse calibrated to perform a 180° rotation at 58 ppm whose duration is set to 53 μs duration to put the first null at 180 ppm on a 14.4 T spectrometer (Supplementary

Note 3.1), and a Q3 180°³ pulse (256 μs at 12.9 kHz with ca. 80 ppm excitation window). Only the Seedless pulse leaves the CO unperturbed and provides a uniform excitation profile of the Cα/Cβ region. **B** HN(CA)CO (**C**, Supplementary Note 3.7) and HNCO (Supplementary Note 3.5) sequences using rectangular (blue) and Seedless (red) pulses were acquired on a U[¹H/¹³C/¹⁵N] ABP1P (**D**, "Methods"). 1D spectra are shown, acquired on a room temperature probe at 600 MHz. Substantial sensitivity gains were obtained using the Seedless pulses. The HNCO spectrum was also recorded using Q pulses as implemented in the Bruker standard library, which shows comparable performance to the sequence with rectangular pulses (Fig. 5).

## Table 3 | Specific ¹³C pulses used within the triple resonance sequences

| Use | Sequences used | Specific pulse | Duration (μs) |
|---|---|---|---|
| **Non-selective excitation:** | | | |
| Excite CO/Cα/Cβ (Iz-Iy) | co, ca, caco, coca | ¹³C CO/Cα/Cβ z-y | 200 |
| **Selective excitation:** | | | |
| Excite CO, hold Cα/Cβ on z | co, caco, coca | ¹³C CO z-y Cα/Cβ zz | 80 |
| Excite Cα/Cβ, hold CO on z | ca, caco, coca | ¹³C CO zz Cα/Cβ z-y | 80 |
| **Selective de-excitation:** | | | |
| De-excite CO, hold Cα/Cβ on z | co, caco, coca | ¹³C CO yz Cα/Cβ zz | 80 |
| De-excite Cα/Cβ, hold CO on z | ca, caco, coca | ¹³C CO zz Cα/Cβ yz | 80 |
| **Invert passive spin (INEPT):** | | | |
| Invert CO, hold Cα/Cβ on z | co, ca, caco, coca | ¹³C CO z-z Cα/Cβ zz | 130 |
| Invert Cα/Cβ, hold CO on z | ca, caco, coca | ¹³C CO zz Cα/Cβ z-z | 130 |
| CO$_x$ → CO$_x$ CA/Cβ$_z$ INEPT | coca | ¹³C CO 180x Cα/Cβ z-z | 200 |
| **Indirect evolution/decoupling:** | | | |
| Detect Cα, decouple CO/Cβ | ca, caco | ¹³C CO/Cβ z-z Cα Id | 400 |
| Detect CO, decouple Cα/Cβ | co, caco | ¹³C CO Id Cα/Cβ z-z | 200 |
| **Cα$_x$ Cα$_y$CO$_z$ INEPT transfer:** | | | |
| Invert Cα/CO, hold Cβ on z | coca | ¹³C CO z-z Cα u180x Cβ zz | 400 |

29 individual pulses were employed in the HNCO, HNCA, HNCOCA and HNCACO sequences, spanning 11 transforms. The HNCA and HNCOCA share a carrier (58 ppm) as do the HNCO and HNCACO (176 ppm), and pulses that share the same transform but have a different carrier need to be calculated separately. The more complex the transform, the longer the total duration required to achieve a reasonable infidelity. The most challenging pulses require independent handling of CO, Cα and Cβ as required for CO/Cβ decoupling during Cα evolution, and Cα/CO INEPT transfer with Cβ decoupling (400 μs in both cases). All these pulses were applied at a peak field of 17.6 kHz, corresponding to the field where a 14.2 μs pulse achieved a 90° pulse, a value obtained from manual calibration on the sample. The pulse transforms used are z-y (S2S excitation), yz (S2S de-excitation), zz ("hold" used for decoupling), z-z ("inversion" used for decoupling and INEPT transfer), 180x (unitary rotation for refocusing) and Id (identity operation, used to pause chemical shift evolution from the perspective of this spin, while others are independently adjusted). See Table 2 for specific instances, Table 1 for the restraints and Supplementary Note 2 for detailed descriptions of the transforms.

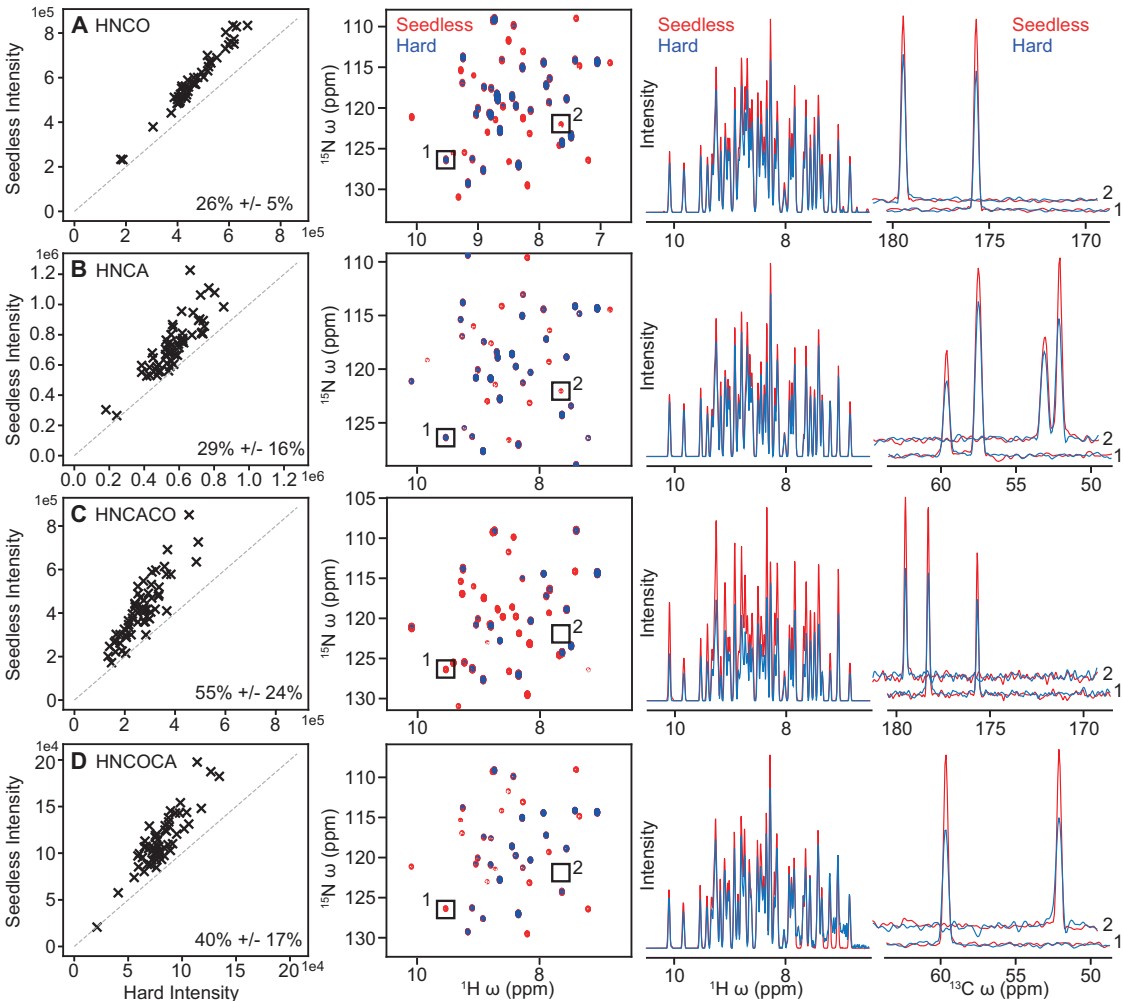

**Fig. 4 | Sensitivity enhancements achieved with triple resonance Seedless pulse sequences.** A comparison of signal intensity between Seedless optimised and rectangular pulse sequences for HNCO (**A**), HNCA (**B**), HNCACO (**C**) and HNCOCA (**D**), described in detail in Supplementary Notes 3.5–8. ¹H/¹⁵N 2D projections, a 1D ¹H skyline projection and indicates slices (1 and 2) are shown. No peaks were found to have lost intensity upon application of Seedless pulses with the intensity of certain residues being doubled.

HSQC, as two pathways need to be controlled requiring universal pulses (Supplementary Note 3.4). For CO detection, a perfectly phased and distortion-less spectrum could be obtained using a central pulse that performs a S2S Z → −Z inversion on Cα/Cβ, and a unitary "identity" operation on the CO spin, avoiding the need for carbon decoupling, with similar approaches used for the Cα.

Finally, we considered coupling between the Cα and Cβ. Because of the substantial overlap of the Cα and Cβ chemical shifts in proteins (Fig. 3A), any attempt to perform different transforms on the Cα and the Cβ will be imperfect, leading to an overlap region where neither Cα nor Cβ are well treated. Because coupling of CO to Cβ is weak, Cα and Cβ can be treated identically, unless one or both is transverse (which happens in two specific situations in these sequences). In the HN(CA)CO, there is a Cα →CO INEPT transfer where Cα is transverse and can couple to the Cβ, resulting in signal loss. To mitigate against this, the central ¹³C pulse needs to perform a unitary 180 rotation of the transverse Cα, a S2S inversion of CO (which together allow Cα→ CO transfer) and a S2S Z→Z "hold" applied to Cβ for decoupling (Section S.3.7). To achieve acceptable performance, the Cα /Cβ interface was placed at 40 ppm. The resulting pulse (400 μs at 17.6 kHz) had an interfacial region of +/− 4 ppm (Supplementary Note 3.6/7). The interface region could be further reduced if the pulse length, spectrometer field or amplitude were increased. This pulse will erode

sensitivity of Cα resonances between 45 and 40 ppm but will completely decouple all Cαs from Cβs where the Cβ was 35 ppm or lower (which for example, automatically excludes all serine and threonine residues, Fig. 3A).

Similarly, in the HNCA and HN(CO)CA sequences during Cα indirect chemical shift acquisition, for distortion-less spectra, there is a need for a pulse that applies an identity operation on Cα, with inversions on both CO and Cβ for decoupling (Supplementary Note 3.6/8). This was also achieved with a 400 μs pulse 17.6 kHz, allowing us to Cβ decouple most amino acid types. We recommend the conventional wisdom of recording spectra at a resolution below which the coupling can be resolved, and so the majority Cβ[28] decoupling manifests as a sensitivity gain (supplementary Fig. 6), though experiments targeting specific residue types could also be constructed[48].

The sensitivity of the resulting pulse sequences was substantially improved in all cases, with sensitivity gains of 26% (HNCO Figs. 3B and 4), 29% (HNCA, Fig. 4), 58% (HNCACO Fig. 3B) and 40% (HNCOCA, Fig. 4), acquired at 600 MHz on a room temperature probe, as judged from comparing the intensity of individual resonances versus an equivalent sequence acquired using rectangular pulses.

To understand more precisely the origin of our sensitivity gains we recorded a series of related HNCO pulse sequences at 600 MHz. At

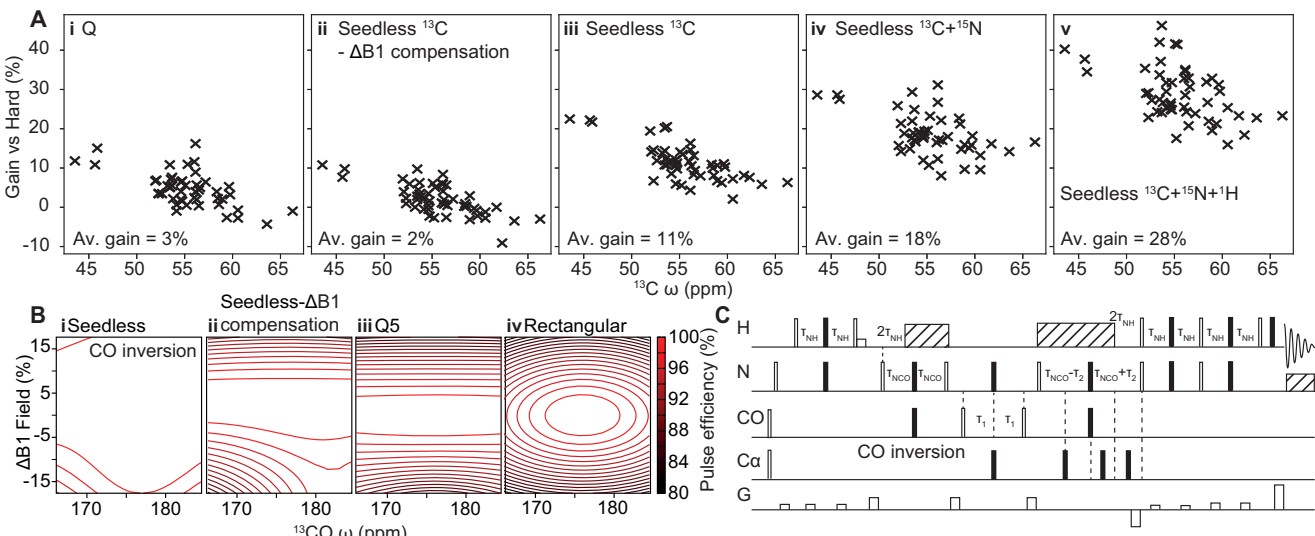

**Fig. 5 | B$_1$ amplitude compensation delivers sensitivity gains.** A detailed comparison of the incremental changes made to an HNCO (pulse sequence **B**, Supplementary Note 3.5) and the variation of sensitivity on Cα chemical shift, as judged on 2D $^{13}$C/$^1$H planes acquired on U[$^1$H/$^{13}$C/$^{15}$N] ABP1P on a room temperature probe at 600 MHz. The number stated is the average sensitivity enhancement in each case. **A i** A pulse sequence from the Bruker standard library (hncocacbgp3d) using Q pulses rather than rectangular pulses behaved similarly, although glycine peaks (ca. 45 ppm) were appreciably enhanced. **A ii**
Non-amplitude compensated $^{13}$C Seedless pulses were included, with intensities like the sequence with Q pulses. **A iii** Introducing inhomogeneity compensated $^{13}$C pulses then $^{15}$N (**A iv**) and finally $^1$H (**A v**) yielded approximately 10% increases in S/N with each nucleus converted. In this case the sensitivity gains are principally derived from the B$_1$ inhomogeneity compensation. **C** The variation of pulse fidelity versus chemical shift for a range of B$_1$ values for a Seedless pulse, pulse (**i**), a Seedless pulse without inhomogeneity compensation (**ii**) and a Q5 pulse (**iii**).

this field, there was no correlation between the sensitivity gains and the $^{13}$C chemical shift, showing that improved excitation bandwidth did not provide our sensitivity. We compared the signal intensities from the sequences with rectangular pulses on $^{15}$N and $^1$H rectangular pulses, and on $^{13}$C we used either rectangular, Q pulses (hncocacbgp3d from the Bruker standard library, Fig. 5Ai), or Seedless pulses, but without B$_1$ compensation (Fig. 5Aii). The sensitivities of all 3 were similar. By including $^{13}$C inhomogeneity compensated pulses (Fig. 5Aiii), then $^{15}$N (Fig. 5Aiv) and then $^1$H (Fig. 5Av), we obtained approximately 10% signal to noise gains per modified nucleus channel. The sensitivity gains are coming purely from B$_1$ compensation, where we are effectively increasing the coil volume and the more pulses in the sequence, the greater the gains. At higher fields, as seen for the $^{15}$N HSQC (Supplementary Fig. 5), we anticipate additional gains coming from both inhomogeneity compensation, and improved performance over a wider range of chemical shifts.

## Discussion

Seedless is a versatile tool for calculating pulses for NMR experiments. Owing to an efficient formulation and implementation of the theory, pulses to handle single spin ½ particles with many restraints can be easily calculated at a rate fast enough for on-the-fly computations. This allows users to easily refine/optimise pulses for specific sample/hardware combinations. Seedless pulses outperform rectangular pulses and shaped pulses that have no homogeneity compensation and provide a means by which bandwidth can be tuned to individual samples (or simply maximised). The inhomogeneity compensation effectively increases the coil volume, providing sensitivity increases that scale with the number of pulses in the sequence. This compensation is not only compensating for potential errors in the specific value used for the central B$_1$ field calibration. Instead the gains reflect the need for the pulse to be effective in all parts of the NMR tube where the amplitude of any B$_1$ field naturally varies, even when the central value has been well calibrated, and that for maximum sensitivity gains a new pulse should be generated reflecting the specific dielectric of the current sample.

We would expect the gains to reflect the specific hardware being used and include factors such as tube geometry that we do not investigate here. We have tested the pulses at 600 MHz and 950 MHz on both room temperature and cryogenically cooled probes using 5 mm NMR tubes. The inhomogeneity distributions were found to be similar in each case. More substantial gains were seen at 950 MHz (58% enhancement of sensitivity enhanced HSQC versus 18% at 600 MHz), reflecting additional substantial gains arising from both improved excitation profiles and homogeneity compensation, with further improvements expected as the field increases further.

We have compared the infidelities of a series of recently produced optimal control pulses against those generated by Seedless under conditions that match (as far as we can ascertain) those originally used. In all cases, Seedless pulses achieve lower infidelities (though in some cases the difference is modest and would be unlikely to show up as a signal to noise gain at the spectrometer, Supplementary Note 5). In cases where the infidelity is comparable under a given set of restraints, by then re-computing the pulses simply to be shorter in biomolecular cases where relaxation losses are significant, we can significantly boost sensitivity (Supplementary Fig. 7). We anticipate that it will always be desirable to match pulses to the specific conditions required by a given hardware/sample combination and that implementing Seedless pulses will be more economical than achieving the same sensitivity gains through purchasing hardware upgrades. As Seedless pulses are generated in a few seconds and the alternative approaches take many hours, we anticipate that Seedless will be a useful general tool for the NMR community.

To create a pulse, a user supplies a peak amplitude (in kHz, determined from a calibrated 90° pulse at the maximum power level), a spectrometer frequency (in MHz), a carrier (in ppm), the number of rectangular elements in the pulse, the total duration of the pulse, and a series of bands that span a specified ppm range for a given number of spins, and an operation to be performed. An inhomogeneity distribution is supplied, and the required pulses are returned together (in batch as needed) with a PDF report showing how the pulses perform (Supplementary Note 4). Usage instructions are supplied ("Methods",

Supplementary Note 4), and all pulses, pulse sequences and input files associated with this manuscript are available for download. More generally, pulses with a longer duration and higher amplitude will lead to more effective pulses with sharper response functions between regions of differing chemical shift. In practical applications, the maximum amplitude will be set by hardware restraints, and the maximum duration will be set by the relaxation rates in the system, and so suitable compromises will need to be found. The one exception to this here is the water selective "suppression" pulses, where there is a need to keep the overall $B_1$ amplitude/duration low so not to disturb water, but still having a reasonably small boundary region between the aliphatic, water and amide bands. Specific and general considerations of how to choose pulses are described both in the text and in the supplementary materials (Supplementary Note 3). The sensitivity of the majority of all routinely used NMR pulse sequences will benefit from the incorporation of these types of pulses.

Seedless is free for academic use, and pre-compiled C++ binaries are available for use compiled under Windows, Linux and macOS.

## Methods

### Pulse sequence design using Seedless

Specific instructions for how to setup a Seedless input script are provided (Supplementary Note 4). The examples described in this work are provided as demonstrations that come with the software download. A report can be automatically generated showing performance of the pulse via three plots showing the fate of magnetisation initially aligned on the *X*, *Y* or *Z* axes, as well as showing how the infidelity falls as a function of iteration during an optimisation. In the following, we consider general principles for making new pulses used in this work. Specific considerations for each featured pulse sequence are described in detail (Supplementary Note 3).

In practice, all pulses are limited by the choice of total duration and $B_1$ amplitude dictated by the spectrometer hardware. Typically, a high $B_1$ amplitude maximises control, which allows the total duration to be minimised, which will reduce losses due to relaxation during the pulse. The only exception in this work is the water suppressed 1D, where lower fields on $^1$H were necessary to minimally perturb water (Fig. 2 and Supplementary Note 3.2). As discussed in the text, it is essential to include a range of $B_1$ fields to account for inhomogeneity distributions. This need is well met by including 3 multiples of the central value, set to 0.93, 1.0 and 1.05 weighed 0.25, 0.5 and 0.25, respectively[26]. We found this well mimics the expected variation in $B_1$ field for a wide variety of probes and spectrometers used for testing this work. This distribution can be tailored to any specific shape as required.

The types of transformations required for a pulse must be carefully considered, and the minimum amount of control should be selected to minimise the pulse duration (shown graphically in Supplementary Fig. 1). This means using S2S, rather than universal restraints where-ever possible. As described in the text, a universal 90°x rotation will perform 3 cardinal rotations of the Bloch sphere, $Z \rightarrow -Y$, $Y \rightarrow Z$, $X \rightarrow X$, and their negated equivalents $-Z \rightarrow Y$, $-Y \rightarrow Z$, $-X \rightarrow -X$. For excitation/de-excitation, we require only $Z \rightarrow -Y$ (excitation) or $Y \rightarrow Z$ (de-excitation), which can be accomplished with S2S transforms requiring shorter durations than an equivalent universal pulse (at constant $B_1$ field). Note that a $Z \rightarrow -Y$ S2S pulse will automatically also perform $-Z \rightarrow Y$, but its behaviour will be uncontrolled if applied to magnetisation initially in the XY plane. Any unitary transformation which performs correctly for two orthogonal rotation axes will also perform correctly for the third, and so there is no reason to consider a "two-axis" case between S2S and unitary pulses. For refocusing pulses during a spin echo or on the active spin during an INEPT, there is no option but to use unitary 180° pulses, as a range of incoming states spread in the XY plane need to be handled, reflecting variable durations of the flanking delays. Within an INEPT transfer, the

passive spin requires a S2S inversion ($Z \rightarrow -Z$) pulse, and so a universal rotation is unnecessary. If a pulse must handle multiple transforms, such as in a sensitivity enhanced refocused INEPT, where we need to simultaneously perform $X \rightarrow X$ and $Z \rightarrow -Y$, then a unitary pulse is essential.

For cases such as the $^{13}$C pulses for triple resonance experiments, we require bands of chemical shift to be treated independently. The ability of the pulse to perform these actions will depend on how different the transformations in the two (or more) bands are, and how separated the bands are in chemical shift. In the triple resonance cases presented, the separation between the Cα and the CO is ca. 80 ppm, which, given the regions themselves are ca. 40 ppm wide, can be considered a substantial gap, allowing near independent control of the two regions. By contrast, the overlap between the Cα and Cβ regions renders independent control over both regions impossible when using single spin methods (Supplementary Fig. 6). A user must make a choice on where the dividing line will be and tolerate that there will be a region adjacent to the dividing line where neither state is well handled, where the longer the duration of the pulse and the higher the spectrometer field, the smaller the compromised overlap region. To overcome this one could conceive of performing multiple experiments, each with a slightly different dividing lines to achieve control or tolerate a compromise.

When introducing a band, the number of frequencies within it must be specified. Seedless distributes these frequencies evenly throughout the band. Too few frequencies, and the performance at intermediate values might be terrible. Too many frequencies, and the computation will take an unnecessarily long time. To find an optimal spacing, we recommend an empirical approach. The Seedless reports show the performance of the pulse. The frequencies specified for this plot should be placed on a different (and more finely spaced) grid than the frequencies used for optimisation. In this way, it becomes obvious in the performance report if a pulse of low infidelity is achieved, but the intermediate frequencies are poorly handled. We recommend ca. 3 or 4 times more points in the performance plot over the optimisation computation. If the performance at intermediate frequencies is poor, simply repeat the optimisation with a finer sampling of frequencies. In general, the optimal pulse will typically perform as expected both on the frequencies used for optimisation and the frequencies in between, but when this need is tensioned against also trying to achieve $B_1$ compensation, increasing the number of frequencies in the computation effectively enhances a user's priority to achieve uniform performance over a frequency range during an optimisation.

Finally, consideration should be given to the duration of the finite element within the pulse. Naively, the duration could be set at the shortest value permitted by the hardware, typically ca. 200 ns, to enable the maximum flexibility in control. These risks excessive transients generated by the electronics at each step, and we see expect improved results by choosing a longer finite element. As a guideline, at maximum power (ca. 25 kHz), a 1 µs finite element applied on resonance will cause a rotation of less than 10°, which we find to be a reasonable target for setting this length. In practical cases, we see only a modest change in overall infidelity upon increasing the length of the finite element, up to this limit (Supplementary Note 5), and so the duration of the finite element can be lengthened without compromising performance.

Convergence of the computation is monitored by following the infidelity versus the iteration number in the report (Supplementary Note 4 and Supplementary Fig. 4). Starting from a random phase (Supplementary Note 2), this decreases rapidly to a plateau, though the time taken and the number of iterations required to achieve this will depend on the specific computation. Values that control convergence of the BFGS optimiser can be altered (Supplementary Note 4), but typically the final optimal value of infidelity is not known in advance, and so we recommend setting and monitoring the maximum

number of iterations, increasing it until the infidelity effectively shows zero variation with iteration. In all practical cases, a range of pulses durations were tested, and the minimum duration was found that gives "good enough" performance was selected, allowing for the formation of intuition. Loosely, if the relevant transformation bands are smooth on the performance graphs over the desired regions of chemical shift, performance on the spectrometer should be excellent. It is not essential to have an infidelity of $10^{-6}$ to have a useful pulse. Typically, infidelity values around $10^{-4}$ were considered tolerable, where differences between values of $10^{-3}$ and $10^{-4}$ often corresponded with noticeable differences in signal to noise. In the specific case of water suppression pulses, where it is impossible to completely avoid exciting the water, higher values were accepted, and the performance was as described.

The final infidelity of a pulse can vary marginally depending on the value of the seed used to compute the random phases that initialise the optimisation. In the examples tested for this work, the variation between the "best" and the "worst" pulse was at most a factor of 3, occurring only when the infidelity is low ($< 10^{-4}$), a value at which it is unlikely to result in a significant sensitivity difference at the spectrometer. Nevertheless, we have included a "horserace" option that allows several computations to be run in parallel, each starting from a different seed, for a user-specified number of iterations. The trajectory with the lowest infidelity at the specified maximum number of iterations is then taken to completion. Having ca. 50 trajectories followed for 100 iterations provides a robust method to reliably obtain pulses that vary only by a small percentage between different starting seed values in the cases described in this work, increasing the computation time by only a modest fraction (ca. 10%). In normal use, we expect this not to be needed, but for a case where the absolute "best" is required, this mode can be used.

## Hardware used for testing

Four spectrometers/probes were used in this work. Similar inhomogeneity profiles were measured in each case, and so all pulses were optimised using the 0.93/1.00/1.05 scheme described in the text unless otherwise stated.

1. A Varian DD2 spectrometer with an Oxford Instruments 600 MHz electromagnet with and a triple resonance probe equipped with XYZ gradients ($^{13}$C HSQC imaging, Fig. 1A and Supplementary Note 3.1).
2. Bruker AVANCE NEO 600 MHz spectrometer with CPRHe-QR-1H/19F/13C/15N-5mm-Z helium-cooled cryoprobe ($^{19}$F 1D, Fig. 1B and Supplementary Note 3.2).
3. A Bruker Avance III HD console with an Oxford Instruments 950 MHz electromagnet and a 5-mm TCI CryoProbe ($^{15}$N HSQC, Fig. 2B and Supplementary Note 3.4)
4. A Varian DD2 spectrometer with an Oxford Instruments 600 MHz electromagnet with 5 mm room temperature HCN probe equipped with Z gradients ($^{13}$C HSQC imaging, Fig. 1A and Supplementary Note 3.1, $^{15}$N HSQC, Fig. 2B and Supplementary Note 3.4, 1H water suppressed 1D, Fig. 2A and Supplementary Note 3.3, HNCO/HNCA/HNCOCA/HNCACO Figs. 3–5 and Supplementary Notes 3.5–8).

## Samples used for testing

The "AutoTest" sample produced by Agilent Technologies was used for the imaging experiments, comprising 99.8% $D_2O$, 0.1% $^{13}$C methanol, 0.1% $^{15}$N Acetonitrile and 0.3 mg/ml $GdCl_3$ (Fig. 1, Supplementary Note 3.1 and Supplementary Fig. 2).

For the $^1$H water suppressed 1D (Fig. 2A and Supplementary Note 3.3), a 10 µM sample of HEWL purchase from Sigma Aldrich was dissolved in PBS at pH 7.4.

U-[$^{15}$N, $^{13}$C] ABP1P Prepared as described previously[42], used to test the $^{15}$N HSQC, HNCO, HNCA, HNCACO and HNCOCA experiments

(Figs. 2–5, Supplementary Notes 3.4–8 and Supplementary Figs. 5–7). The buffer used was 50 mM NaPi, 100 mM NaCl, 2 mM EDTA, 2 mM $NaN_3$, 10% $D_2O$ pH 7.

For the $^{19}$F experiment (Fig. 1, Supplementary Note 3.2 and Supplementary Fig. 3), Selectfluor (containing two $^{19}$F containing species, Sigma), and Fmoc-L-MfeGly were mixed in a 1:1 molar ratio in deuterated acetonitrile. The expected $^{19}$F chemical shifts span a wide range, at 47.89 ppm, −151.5 ppm and −221.5 ppm, respectively. Fmoc-L-MfeGly (final compound, below) was synthesized in the following 5 step scheme described in detail below, following a method described previously[49].

### *tert*-Butyl N-(*tert*-butoxycarbonyl)-ʟ-homoserinate (Boc-ʟ-hSe-O*t*Bu)

Under argon, Boc-L-Asp-O*t*Bu (5.00 g, 17.3 mmol) was dissolved in dry THF (170 mL) and the solution was cooled to 0 °C. Isobutyl chloroformate (6.72 mL, 51.8 mmol) and DIPEA (4.52 mL, 25.9 mmol) were added and the resulting mixture was allowed to stir at 0 °C for 1 h before warming to RT and stirring for a further 30 min. Sodium borohydride (4.58 g, 121 mmol) was added slowly at 0 °C followed by $H_2O$ (40 mL) under a stream of nitrogen. The mixture was left to stir overnight at RT and then acidified with 1 M HCl until pH 2. The organic layer was separated and the aqueous phase was extracted with EtOAc (3 × 100 mL). The combined organic layers were washed with sat. aq. $NaHCO_3$ (2 × 100 mL) and brine (100 mL) then dried over $MgSO_4$, filtered and concentrated under reduced pressure. Combiflash purification by silica gel column chromatography (0–80% EtOAc in pet. ether over 18 min) gave Boc-L-Hse(homoserine)-O*t*Bu as a colourless oil (4.36 g, 92%).

$C_{13}H_{25}NO_5$ (275.4 g/mol): $^1$H NMR (400 MHz, $CDCl_3$) δ 5.36 (br s, 1H), 4.38–4.30 (m, 1H), 3.72–3.59 (m, 3H), 2.17–2.08 (m, 1H), 1.56–1.49 (m, 1H), 1.46 (s, 9H), 1.44 (s, 9H). $^{13}$C NMR (101 MHz, $CDCl_3$) δ 172.2, 156.8, 82.4, 80.5, 58.4, 51.0, 36.7, 28.4 (3 C), 28.1 (3 C). Spectroscopic data was consistent with literature reports[49].

### *tert*-Butyl N-(*tert*-butoxycarbonyl)-O-tosyl-ʟ-homoserinate (Boc-ʟ-hSe(OTs)-O*t*Bu)

The following procedure was adapted from a known procedure[49]. Under argon, Boc-L-Hse-O*t*Bu (825 mg, 3.00 mmol) was dissolved in dry DCM (9 mL) and the solution was cooled to 0 °C. NEt$_3$ (2.09 mL, 15.0 mmol), TsCl (1.14 g, 5.99 mmol) and DMAP (36.6 mg, 0.300 mmol) were added sequentially and the resulting mixture was allowed to stir at 0 °C for 15 min before warming to RT and stirring for a further 16 h. The mixture was then diluted with DCM (12 mL) and the organic phase was washed with $H_2O$ (2 × 20 mL) and brine (10 mL), then dried over $MgSO_4$, filtered and concentrated under reduced pressure. Combiflash purification by silica gel column chromatography (5 to 15% EtOAc in pet. ether over 15 min) gave Boc-L-Hse(OTs)-O*t*Bu as a white solid (921 mg, 72%).

$C_{20}H_{31}NO_7S$ (429.5 g/mol): $^1$H NMR (400 MHz, $CDCl_3$) δ 7.78 (d, $J = 8.4$ Hz, 2H), 7.34 (d, $J = 8.2$ Hz, 2H), 5.01 (d, $J = 5.9$ Hz, 1H), 4.17–4.15 (m, 1H), 4.08 (td, $J = 6.5$, 2.1 Hz, 2H), 2.44 (s, 3H), 2.23–2.18 (m, 1H), 2.08–1.99 (m, 1H), 1.44 (s, 9H), 1.40 (s, 9H). $^{13}$C NMR (101 MHz, $CDCl_3$) δ 170.7, 155.3, 145.0, 132.9, 130.0 (2 C), 128.2 (2 C), 82.8, 80.0, 66.7, 51.2, 31.8, 28.4 (3 C), 28.0 (3 C), 21.8. MS$^+$ 430.1 [M + H]$^+$, 452.2 [M + Na]$^+$ Alpha D $[\alpha]_D^{25} = -16.5$ (c = 1.0, MeOH). Spectroscopic data was consistent with literature reports[49].

### *tert*-Butyl (S)-2-(*tert*-butoxycarbonyl)amino)-4-fluorobutanoate (Boc-ʟ-MfeGly-O*t*Bu)

The following procedure was adapted from a known procedure[49]. Under argon, Boc-L-Hse(OTs)-O*t*Bu (847 mg, 1.97 mmol) was dissolved in anhydrous *t*BuOH (20 mL). TBAF·3H$_2$O (1.87 g, 5.92 mmol) was added and the resulting mixture was allowed to stir at 70 °C for 3 h before quenching with sat. aq. $NaHCO_3$ (20 mL). The mixture was

extracted with DCM ($3 \times 30$ mL) and the combined organic phases were dried over $MgSO_4$, filtered and concentrated under reduced pressure. Combiflash purification by silica gel column chromatography (0 to 20% EtOAc in pet. ether over 13 min) gave Boc-L-MfeGly-OtBu as a white solid (397 mg, 73%).

$C_{13}H_{24}FNO_4$ (277.3 g/mol): $^1$H NMR (400 MHz, $CDCl_3$) δ 5.21 (d, $J = 6.2$ Hz, 1H), 4.64–4.55 (m, 1H), 4.52–4.43 (m, 1H), 4.34–4.24 (m, 1H), 2.31–1.96 (m, 2H), 1.47 (s, 9H), 1.44 (s, 9H). $^{19}$F NMR (376 MHz, $CDCl_3$) δ −219.4 (m). $^{13}$C NMR (101 MHz, $CDCl_3$) δ 171.1, 155.5, 82.4, 80.6 (d, $J = 166$ Hz), 80.0, 51.3, 33.5 (d, $J = 19.8$ Hz), 28.4 (3 C), 28.1 (3 C). $MS^+$ 278.4 $[M + H]^+$. Spectroscopic data was consistent with literature reports[49].

### (S)-2-amino-4-fluorobutanoic acid trifluoroacetate (L-MfeGly•CF₃CO₂H)

A solution of Boc-L-MfeGly-OtBu (386 mg, 1.39 mmol) in $H_2O$ (1.4 mL) was cooled to 0 °C. TFA (12.5 mL) was added, and the reaction was allowed to stir at RT for 4 h. The solution was concentrated under a stream of nitrogen and dried azeotropically under reduced pressure with toluene to give the TFA salt of L-MfeGly-OH as a colourless solid (quant. yield). The unprotected amino acid was used directly in the next step.

### (S)-2-((((9H-fluoren-9-yl)methoxy)carbonyl)amino)-4-fluorobutanoic acid (Fmoc-L-MfeGly)

N-(9-Fluorenylmethoxycarbonyloxy)succinimide (FmocOSu) (493 mg, 1.46 mmol) was added to a mixture of crude L-MfeGly-OH (1.39 mmol) in THF (10 mL) and sat. aq. $NaHCO_3$ (5 mL). The resulting mixture was stirred at RT for 16 h and then diluted with $H_2O$ (10 mL). The aqueous layer was washed with $Et_2O$ ($2 \times 25$ mL) and acidified with 1 M HCl until pH 1–2. The aqueous phased was extracted with EtOAc ($3 \times 30$ mL). The combined organic layers were dried over $MgSO_4$, filtered and concentrated under reduced pressure to give Fmoc-L-MfeGly-OH as a white solid (414 mg, 82% by $^1$H NMR) without further purification.

$C_{19}H_{18}FNO_4$ (343.35 g/mol): $^1$H NMR (600 MHz, Acetone-$d_6$) δ 7.86 (d, $J = 7.6$ Hz, 2H), 7.72 (t, $J = 6.6$ Hz, 2H), 7.41 (t, $J = 7.5$ Hz, 2H), 7.32 (td, $J = 7.4, 1.2$ Hz, 2H), 6.83 (d, $J = 8.5$ Hz, 1H), 6.83 (d, $J = 8.6$, 1H), 4.68–4.62 (m, 1H), 4.60–4.54 (m, 1H), 4.43–4.38 (m, 1H), 4.37–4.33 (m, 2H), 4.25 (t, $J = 7.2$ Hz, 1H), 2.39 – 2.30 (m, 1H), 2.17–2.09 (m, 1H). $^{19}$F NMR (565 MHz, Acetone-$d_6$) δ −221.5 (tdd, $J = 47.3, 29.6, 21.0$ Hz). $^{13}$C NMR (151 MHz, Acetone-$d_6$) δ 173.5, 157.1, 145.1 (2 C), 142.1 (2 C), 128.5 (2 C), 127.9 (2 C), 126.2 (2 C), 120.8 (2 C), 81.2 (d, $J = 164.0$ Hz), 67.2, 51.3 (d, $J = 4.9$ Hz), 48.0, 33.2 (d, $J = 20.3$ Hz). $MS^-$ 342.1 $[M - H]^-$ Alpha D $[\alpha]_D^{25} = -21.4$ (c = 1.0, MeOH). Spectroscopic data was consistent with literature reports[49].

### Specific Seedless pulses

In this work, 8 pulse sequences were analysed and executed on three different spectrometers and four probes, requiring 54 bespoke Seedless pulses (Table 2). All pulses can be computed using the demonstration scripts provided with the software download. Considerations and settings used are described below, and detailed descriptions of each pulse sequence are provided (Supplementary Note 3).

For the $^{13}$C HSQC imaging, $^{19}$F 1D sequences and $^{15}$N HSQC (Supplementary Notes 3.1, 2, 4), we required a single transform over a range of chemical shift. In all other cases (1D water suppression, HNCO, HNCA, HNCOCA, HNCACO Supplementary Notes 3.3, 5–8), different regions of chemical shift were handled independently. Typically, lengthening the pulse leads to a "sharper" and more desirable transition between the interfaces, but the total length of the pulse cannot exceed hardware limits on the total applied power. Also, a longer duration pulse leads to greater relaxation losses, so all pulses represent "minimally bad" compromises.

In cases where simultaneous excitation over multiple nuclei is required, the durations of the two pulses was synchronised. Typically,

this means one pulse has an "easy" task to perform (e.g., non-band selective, high gamma $^1$H) and the other has a "hard" task (e.g., band selective, low gamma $^{13}$C), so in principle the time of the "easy" pulse, or its amplitude could have been reduced. Because of the extra time available, we tended to apply universal operations on the "easy" spin for convenience (usually on $^1$H) rather than working with specific S2S variants. This means that all pulses in isolation or synchronized can be considered "zero-time" removing the need to subtract any delays to balance the effects of the pulse. This both simplifies implementation and leaves indirect dimensions perfectly phased.

**$^{19}$F pulses.** For the 300 ppm broadband excitation pulse, a 2000 μs pulse was created at a field of 20.2 kHz using 300 frequencies spaced evenly over this range, corresponding to a field where a rectangular pulse requires 12 μs to execute a 90° rotation (Supplementary Note 3.1).

**$^1$H pulses.** The carrier was set to be 4.77 ppm water. All pulses (HSQC and triple resonance) were designed for a uniform bandwidth (0–12 ppm with 96 evenly spaced frequencies) unless otherwise stated (Supplementary Note 3.4). For the water selective pulse sequence, the aliphatic band (CH) and amide (NH) band were defined as −1 to 3.5 ppm (15 frequencies), 6.5 ppm to 10 ppm (20 frequencies) and water band 4.65 to 4.85 ppm (10 frequencies) respectively (Supplementary Note 3.3). The peak field was 25 kHz, corresponding to a field where a rectangular pulse requires 10 μs to execute a 90° rotation. Pulse durations ranged from 100 to 250 μs for non-band-selective pulses at 17.6 kHz, and 2000–4000 μs for the suppression pulses at 9.76 kHz.

**$^{15}$N pulses.** A specific region from 106 to 133 ppm was chosen that spans the range of chemical shifts observed for abp1p (60 evenly spaced frequencies), and the carrier set to 119.5 ppm. The peak field was 6.85 kHz, corresponding to a field where a rectangular pulse requires 36 μs to execute a 90° rotation. Pulse durations were set to 250 μs.

**$^{13}$C pulses.** For the triple resonance applications, three bands were defined as discussed in the text for CO, Cα and Cβ, with Cβ from 8 to 40 ppm, Cα from 40 ppm to 78 and CO from 158 to 198 ppm. The types of transformations across the triple resonance experiments can be summarised in 7 different classes (Table 3). The carriers for the HNCA/HNCOCA and HNCO/HNCACO experiments were set at 58 and 176 ppm, respectively (centred on the required $^{13}$C indirect dimension), and so even if two frequency bands require the same transformation in an HNCO and HNCA, this requires two separate calculations centred at the two different carriers. The peak field was 17.6 kHz corresponding to a field where a rectangular pulse requires 14 μs to execute a 90° rotation. Pulse durations were in the range 80–400 μs. Setting 96 frequencies in each of the Cα, Cβ and CO bands typically gave excellent performance, corresponding to a ~ 0.4 ppm resolution. Having pulses that distinguish CαCβ is demanding. For most $^{13}$C, pulses Cα/Cβ can be safely treated one group. There are two important cases where all three bands must be handled independently. Firstly, in the HNCA and HNCOCA, during the Cα indirect evolution period, we need to decouple transverse Cα from Cβ and CO. We accomplish this with a pulse that performs an identity operation on Cα, effectively pausing indirect frequency evolution while inverting Cβ and CO ($Z \to -Z$). This is a demanding pulse and so requires a longer duration of 400 μs to achieve an acceptable infidelity. Secondly, Cα/Cβ distinction is required is during the Cα $\to$ CO INEPT transfer in the HNCACO. In this case, we have transverse Cα magnetization that will appreciably couple to both Cβ and CO. In this case we need a unitary 180° rotation on the transverse Cα, a $Z \to -Z$ inversion on CO and a $Z \to Z$ transform on Cβ to leave it unchanged. We again achieve this with a 400 μs pulse. When computing pulses that distinguish Cα and

Cβ, we set the number of frequencies on Cα/CO to 96, and for Cβ, 20. This effectively decreases the "weight" of the calculation for the Cβs, giving preference to Cαs.

## Comparisons between Triple resonance sequences

For the 3D HNCO, HNCA, HNCOCA and HNCACO (Supplementary Notes 3.5–8), $^{15}$N and $^{1}$H pulses were used as for the HSQC (Supplementary Note 3.4). The performance of the Seedless modified sequences were compared to triple resonance sequences using rectangular pulses in sequences implemented as they were originally described[46], where 90°/180° rectangular pulses are constructed to apply the desired rotation on resonance, but with durations $t_{90} = \frac{\sqrt{15}}{4\nu}$ or $t_{180} = \frac{\sqrt{3}}{2\nu}$ to ensure their first excitation null will occur at frequencies $\pm \nu$ Hz from the carrier to independently control Cα and CO. In the case of the HNCO sequence, we also compare signal from the original pulse sequence constructed using rectangular pulses, the Seedless pulse sequence, and one constructed using "Q" pulses (Fig. 5), present in the Bruker standard library "hncocacbgp3d", (shown explicitly in Supplementary Section 3.5). The pulse sequences constructed using rectangular pulses and with the Q pulses have explicit Cα decoupling. We accomplish this using single Cα 180° Seedless pulses as shown (Supplementary Note 3.2).

## Data availability

Data used to construct the figures is available for download from https://doi.org/10.5281/zenodo.14227532.

## Code availability

Software is available for download from http://seedless.chem.ox.ac.uk. It is written in C++ and will be available in a pre-compiled binary form. This comes with demonstration scripts that once run will compute pulses described in this manuscript. These can be used as templates for new applications, in conjunction with the usage instructions (Supplementary Note 4). The Software is distributed "AS IS" under this Licence solely for non-commercial use. If you are interested in using the Software commercially, please contact the technology transfer company of the University, to negotiate a licence. Contact details are: "enquiries@innovation.ox.ac.uk". We will thoroughly welcome community input, so positive and negative feedback will be appreciated.

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

## Acknowledgements

We thank Steffen Glaser and Burkhard Luy for helpful discussions, and to colleagues at Bruker, Philip Wurm, Maksim Mayzel and Niels Karschin, for discussion and testing. Thanks to Jesse Schelfhout and Suzanne Lim for support with compiling the software under Windows, and to Bowen Guo, Jack Bercovici, Matthew Davy and Abi Turner for testing. A.J.B. has received funding from the European Research Council (ERC) under the European Union's Horizon 2020 research and innovation programme (grant agreement No 101002859).

## Author contributions

Materials and raw data: G.K., V.C., A.W.J.P., B.G.D, C.J.B., G.B.. Algorithm implementation, development, benchmarking and testing: C.J.B., G.B., J.J., A.J.B. The manuscript was written by C.J.B., J.J. and A.J.B. with input from all authors.

## Competing interests

All authors declare no competing interests.
