## [Transparent Peer Review file · Nature Communications]

Seedless: On-the-fly pulse calculation for NMR experiments

Corresponding Author: Professor Andrew Baldwin

Version 0:

Reviewer comments:

Reviewer #1

(Remarks to the Author)

The manuscript presents a new software for designing NMR radiofrequency (rf) pulses to efficiently manipulate nuclear spins. In the theoretical sections (both the main text and the supplementary), the authors try to bridge two closely related scientific fields of quantum information and magnetic resonance, which I find very valuable. The topic of designing high-performance rf pulses has a long history in NMR, including algorithms based on optimal control theory. Recently, it has been revived in relation to ultra-high magnetic field NMR spectrometers where the rf hardware is challenged by increased frequency bandwidths that need to be covered by rf pulses. Seedless, the new software, implements well known optimal control approaches (might be better cited though). Compared to previous implementations and reports, it benefits from streamlined numerical operations that are based on restricting assumptions that other software don't have. The words "on-the-fly" in the title of this manuscript probably refer to the calculation speed once the user masters all parameters relevant for the pulse design. It is still an art to quickly come to good results and a steep learning curve for a novice. There is also an important factor that modern computers are much more powerful than those used in previous studies. The authors document very well how their software can be useful in almost every NMR application and provide extensive examples. In general, the manuscript is well done and deserves publication after addressing the following comments.

Referencing previous OC applications in NMR could be expanded. The very similar work of Joseph and Griesinger (DOI:10.1126/sciadv.adj1133) is mentioned in the Supplementary but, in my opinion, is very relevant in the main text as well. Both papers report sensitivity improvements of a few tens of percent but in solid-state NMR applications, gains on the order of 100% were recently reported (DOI: 10.1002/anie.201805002). OC implementation in SIMPSON software package is reported in DOI:10.1016/j.jmr.2008.11.020. Principles of the proposed "novel type of pulse" design for water suppression were described earlier by Glaser and dubbed as optimal tracking (DOI:10.1016/j.jmr.2009.07.024).

Seedless pulses are generated assuming an isolated single spin system. It does not account for J-couplings that could be large especially for ^{19}F containing molecules. Likewise, Bloch-Siegert shifts are not discussed while they are important in multidimensional bio-NMR experiments (HNCACO,...) that include special compensation pulses. Possible consequences of ignoring these effects should be discussed in the paper (or explain why it is safe to ignore them).

Table 1 is included both in the main text and in the Supplementary. However, it contains inconsistent information. In addition, it is not described what "Axes" means, especially when it is assigned the value "<1" – what it means? In the caption, we can read "universal rotation requires either an axis and angle or..." – so why „Axes“ equals 3?

Pulse transients – "to mitigate against this we focused on constant amplitude 'phase-only' pulses". That is not sufficient at all, rapid jumps in phases of otherwise constant rf amplitude sequence induce strong rf amplitude transients as well (for example, DOI: 10.1016/j.ssnmr.2012.02.009). More careful discussion is needed here...

I find confusing the term "number of spins in simulation". Seedless is strictly single-spin. "Number of spins" refers to an ensemble of single-spin systems with different characteristics, not multiple-spin calculations. This comes up at different places in the manuscript and the Supplementary.

Figure 1 – Outcome of ^{13}C HSQC imaging are ^1H spectra (panels A), this is quite confusing. This sequence is described in Supplementary and contains multiple rf pulses. It is not clear if the observed gains quantify the cumulative effect when all the pulses are replaced by Seedless?

While the Figures look good and contain lot of information, captions tend to be long and sometimes are missing descriptions of all the panels. Specifically, in Fig. 1 panel Bii is not described, in Fig.2 panel Biv, in Fig. 5 panels B and C. In Fig. 3, the information in the caption is not consistent with the figures.

When quantifying sensitivity gains, it is always difficult what the reference experiment should be. The authors take rectangular pulses. OK, but then it should not be called "Bruker standard". For example, "Bruker standard" for our 600MHz spectrometer is based on chirp pulses and not simple rectangulars...

Water suppression example – the suppressed region is quite large, 3.5-6.5ppm, and might contain important resonances. This should be more stressed in the text. Fair comparison would be with a sequence that has the same blind region (Figure 2 A iv and v)

The sentence „Following a process of largely trial and error ...“ does not sound very scientific. One would guess that for experiments known for decades, the effect of each pulse is already known. And it is known to the authors as well, as they explain nicely in the Supplementary.

Very little is written about how to choose number of offsets representing ppm ranges, number of elements in the Seedless pulse, its duration, width of interface regions, relative weights of simultaneously optimized criteria (regions with different aim) etc. Understanding the interplay among these parameters constitutes "the art of optimization". Unexperienced user will never be optimizing pulses "on-the-fly"...

Minor issues:

„In practical applications, a nucleus, peak field strength, amplitude (peak B1 field),...“ – repetition in the last two terms?

Seedless is sometimes typeset with small „s“ (seedless/Seedless)

“...the need to account for undesirable uncontrolled evolution of chemical shift during pulses during delays ...“ - repetition

Why three levels of referencing in Figures? (Fig. 2 A ii)

Please read the Supplementary carefully again. There are many mistakes and typos...

(Remarks on code availability)

Reviewer #2

(Remarks to the Author)

Andrew Baldwin and coworkers published a remarkable idea for optimal NMR spectroscopy.

It's claim that the parameter optimization is faster is not entirely clear to me and I have to trust. It is also unclear what the difference is to GRAPE's mathematical procedure. I did not find a discussion of convergence criteria. Pulses that are easy and fast to optimize but don't converge are potentially problematic in my experience. Particularly so if they are done on the fly. I generally trust that they tested the procedure.

I generally like that research and hope it lives up to its promises.

(Remarks on code availability)

Version 1:

Reviewer comments:

Reviewer #1

(Remarks to the Author)

The authors adequately addressed the issues raised in the original review, clarified most points, and I respect their experience in others. I fully support publishing this work and I wish the authors good luck with NMR vendors, your software comes out at the right time.

(Remarks on code availability)

Reviewer #2

(Remarks to the Author)

Andrew Baldwin and coworkers' research report about in silico optimized pulses for optimal NMR analysis improved significantly following the reviews.

The work is highly relevant and merits publication.

(Remarks on code availability)

REVIEWER COMMENTS

Reviewer #1 (Remarks to the Author):

The manuscript presents a new software for designing NMR radiofrequency (rf) pulses to efficiently manipulate nuclear spins. In the theoretical sections (both the main text and the supplementary), the authors try to bridge two closely related scientific fields of quantum information and magnetic resonance, which I find very valuable. The topic of designing high-performance rf pulses has a long history in NMR, including algorithms based on optimal control theory. Recently, it has been revived in relation to ultra-high magnetic field NMR spectrometers where the rf hardware is challenged by increased frequency bandwidths that need to be covered by rf pulses. Seedless, the new software, implements well known optimal control approaches (might be better cited though). Compared to previous implementations and reports, it benefits from streamlined numerical operations that are based on restricting assumptions that other software don't have. The words "on-the-fly" in the title of this manuscript probably refer to the calculation speed once the user masters all parameters relevant for the pulse design. It is still an art to quickly come to good results and a steep learning curve for a novice.

We thank the reviewer. We would respectfully disagree with the challenge presented by this type of optimisation. Historically this has been the case. But the implementation we provide here, Seedless, is extremely straightforward to use. A user will need to make certain decisions and know what type of pulse they want to calculate (this of course is unavoidable). But pragmatic aspects such as the number of spins and the optimisation itself, using the 'phase only' pulses make the optimisation very smooth, so there is no problem with local minima. Initialising the calculation in the way that we do, we have for all pulses in the manuscript not once gotten stuck in a local maximum.

To help make things easy to use and to address these concerns, we have expanded section S3 in the supplementary to provide a practical guide for to think about the various design aspects, the number of frequencies to be included in the calculation, the duration of the finite element, the total duration and the B1 amplitudes/inhomogeneity corrections. In the software download we have provided example scripts for all the pulses shown in the manuscript.

There is also an important factor that modern computers are much more powerful than those used in previous studies. The authors document very well how their software can be useful in almost every NMR application and provide extensive examples. In general, the manuscript is well done and deserves publication after addressing the following comments.

We thank the reviewer for their kind words.

Referencing previous OC applications in NMR could be expanded.

We thank the reviewer for raising this. We agree that we should provide a more thorough review of the successes of the field, and we have expanded our introduction taking into account the points below. Overall, we are aiming here to focus on papers that show the fundamental principles, selecting examples to illustrate the major points. It is not within the scope of the manuscript to provide a complete review of all applications of optimal control pulses within NMR.

The very similar work of Joseph and Griesinger (DOI:10.1126/sciadv.adj1133) is mentioned in the Supplementary but, in my opinion, is very relevant in the main text as well.

We now have cited this paper in the introduction.

We would like to stress that the focus of the paper above is to generate a collection of OC pulses, dedicated to 1.2GHz. These were calculated using the program spinach and each takes 12 hours using the program Spinach. In discussions with the Spinach author, Prof. Ilya Kuprov, he has noted that while he is very interested in the general problem of spin physics and far more complex things can be accomplished with OC pulses using his platform, our approach being dedicated to the specific problem of spins $\frac{1}{2}$, we will necessarily have optimal performance characteristics. Thus our calculations for functionally identical pulses take a few seconds (Supplementary Section S6).

To illustrate why this is useful we have further produced supplementary figure 7. We compare the performance of just the ^{15}N pulses in the above paper together with ones we produce. We show that a pulse created using Seedless performs identically to the OC1.2 pulses. But optimising for our hardware, we can reduce the length of the pulse and immediately obtain 15% more signal.

Moreover, it would not be possible, from the pulses made available from the above calculation, to create pulse sequences for the 4 triple resonance experiments as shown in this paper.

While we do agree that our software produces pulses that are very similar both in terms of fidelity to those described in this work, our paper describes a new approach that allows users to create any pulse, including those similar to published, but also all others in a flexible manner that enables, for the first time, a true 'on-the-fly' pulse approach.

Both papers report sensitivity improvements of a few tens of percent but in solid-state NMR applications, gains on the order of 100% were recently reported (DOI: 10.1002/anie.201805002).

We have cited this paper in the introduction.

OC implementation in SIMPSON software package is reported in DOI:10.1016/j.jmr.2008.11.020 .

We have cited this paper in the introduction.

Principles of the proposed "novel type of pulse" design for water suppression were described earlier by Glaser and dubbed as optimal tracking (DOI:10.1016/j.jmr.2009.07.024).

We have cited this paper in our discussion on water suppression. The optimal tracking as discussed in this work is for heteronuclear decoupling in a two spin system. Using this approach for water suppression in a single spin context is a different application and so we stand by our description that this is novel. But we agree that it is appropriate to cite similar work and we will add this into our discussion when introducing our water suppression pulse.

Seedless pulses are generated assuming an isolated single spin system. It does not account for J-couplings that could be large especially for ^{19}F containing molecules. Likewise, Bloch-Siegert shifts are not discussed while they are important in multidimensional bio-NMR experiments (HNCACO,..) that include special compensation pulses. Possible consequences of ignoring these effects should be discussed in the paper (or explain why it is safe to ignore them).

We have added the following into the discussion:

From the perspective of single spins, Bloch-Siegert shifts arising from excitation far off-resonance are automatically included in the calculation and there is no need to make any separate correction for such effects. If pulses are applied to one nucleus at a time then heteronuclear scalar coupling is indistinguishable from chemical shift and is controlled in the same way. Homonuclear scalar coupling can in principle complicate the performance of the pulses, as can heteronuclear coupling when we pulse on both nuclei concurrently, typically observable as a modest alteration of multiplet intensities. However as long as the pulse durations are much shorter than $1/J$, as is normally the case, there is insufficient time

for significant evolution. Numerical simulations for the 2 ms ^{19}F broadband excitation pulse show that the performance of the pulse is close to perfect in the presence of scalar couplings below 200Hz, and acceptable for couplings up to 500Hz.

We have included the following specific discussion in the supplementary information:

“a more careful analysis shows that the main effect of this is to produce a very small Bloch-Siegert frequency shift⁵, subtly changing the offset frequency while the RF is applied. Note that it is only the true Bloch-Siegert shift, arising from the counter rotating component, which is neglected: the closely related phase shift arising from excitation far off-resonance, described by Ramsey, is automatically included⁶.”

Table 1 is included both in the main text and in the Supplementary. However, it contains inconsistent information. In addition, it is not described what “Axes” means, especially when it is assigned the value “<1” – what it means? In the caption, we can read “universal rotation requires either an axis and angle or...” – so why „Axes“ equals 3?

We have clarified this in the resubmission in the legend for the table. In brief we are referring to the three axes of the Bloch Sphere. The Universal rotation controls all points on the sphere and can be described as controlling 3 axes. A state-to-state will control only one axis but not the other two. Thus while, for example, a state-to-state pulse may perform Z->Y, the two accompanying rotations X->X and -Y->-Z were not imposed and so are unlikely to be performed by this pulse. The XYcite pulse, because it does not aim to place precisely where the spin is in the XY plane, it is controlling magnetisation less well than a state-to-state, and so in some sense tries to control fewer than 1 axes of the Bloch sphere, over a band of chemical shifts. We find this to be an intuitive and useful way to think about the pulses, as we discuss in the text, where the more control is required, the longer time / higher B1 fields are required, and if in an application one can get away with less control, this is desirable.

Pulse transients – “to mitigate against this we focused on constant amplitude ‘phase-only’ pulses”. That is not sufficient at all, rapid jumps in phases of otherwise constant rf amplitude sequence induce strong rf amplitude transients as well (for example, DOI: 10.1016/j.snmr.2012.02.009). More careful discussion is needed here...

We have now cited this paper. We agree that transients are expected when we vary either amplitude or phase. As we say in the text, we mitigate against this by holding one of these two constant. We did not say that we have reduced these to zero.

As discussed in the supplementary information, and has now been expanded (section S3), we discuss in more detail the beneficial consequence of keeping the finite element in the pulse to be relatively long. For example the finite elements in the OC pulses in the recent Griesinger paper are 200 ns, but we show we can achieve similar fidelities using a 1 μs finite element. As we note, we would expect this to reduce transients in the coil. Detailed measurements of this are outside the scope of the current work. In the guidelines we recommend making the finite element as long as possible to achieve a desired fidelity.

I find confusing the term “number of spins in simulation”. Seedless is strictly single-spin. “Number of spins” refers to an ensemble of single-spin systems with different characteristics, not multiple-spin calculations. This comes up at different places in the manuscript and the Supplementary.

We agree that our previous statements were ambiguous. To be more explicit we now refer to the ‘number of frequencies’ included in a calculation. We agree that strictly we are considering the action of an ensemble of isolated spins $\frac{1}{2}$ whose chemical shifts are defined in the manner we describe.

Figure 1 – Outcome of ^{13}C HSQC imaging are ^1H spectra (panels A), this is quite confusing. This sequence is described in Supplementary and contains multiple rf pulses. It is not clear if the observed gains quantify the cumulative effect when all the pulses are replaced by Seedless?

We have clarified this in the text. Supplementary figure S2 and Figure 5 help illustrate the origin of signal gains. A conventional HSQC pulse sequence with rectangular pulses gives the same intensity as the same but with pulses replaced by Seedless, but without B1 inhomogeneity compensation. When we included Seedless pulses with B1 inhomogeneity compensation, this is the requirement for us to see gain in signal.

As further discussed in the manuscript with the HNCOs in figure 5, we see roughly 10% gains per channel (^1H , ^{13}C , ^{15}N), giving us an overall gain of 30% overall for the HNCO. We have clarified this in the discussion.

While the Figures look good and contain lot of information, captions tend to be long and sometimes are missing descriptions of all the panels. Specifically, in Fig. 1 panel Bii is not described, in Fig.2 panel Biv, in Fig. 5 panels B and C. In Fig. 3, the information in the caption is not consistent with the figures.

We have clarified this.

When quantifying sensitivity gains, it is always difficult what the reference experiment should be. The authors take rectangular pulses. OK, but then it should not be called “Bruker standard”. For example, “Bruker standard” for our 600MHz spectrometer is based on chirp pulses and not simple rectangulars...

We have clarified this. The sequences using rectangular pulses we have identified by citation and additional description in the SI. We use one pulse sequence from the Bruker standard library, an HNCQ using Q pulses. To be very specific we have included its name (**hncocacbgp3d'**), written out the pulse sequence formally in section S.3.2.5 and given all associated details.

The major purpose of this sequence, in figure 5, is showing that the performance of the rectangular pulses and the Q pulses are quite similar, with sensitivity gains in the HNCQ coming from the B1 compensation present in the Seedless pulses.

Water suppression example – the suppressed region is quite large, 3.5-6.5ppm, and might contain important resonances. This should be more stressed in the text. Fair comparison would be with a sequence that has the same blind region (Figure 2 A iv and v)

We have clarified this in the text. In practical cases users can design gap as they like. A region in this vicinity is always destroyed using water suppression techniques.

“This comes at the expense of removing the indicated region in the vicinity of water from the spectrum (A iv, v).”

The sentence „Following a process of largely trial and error ...“ does not sound very scientific. One would guess that for experiments known for decades, the effect of each pulse is already known. And it is known to the authors as well, as they explain nicely in the Supplementary.

We would disagree and note that this follows exactly our collective experience of NMR accumulated over many decades. The point we wish to make is basically that while we possibly could have figured this all out at the beginning, the actual action of finding which pulses were best in which position was achieved as we describe, through trial and error. This was hard won knowledge.

We can further illustrate this with an anecdote. The project began because we wanted to make pulses that control spins in frequency bands, much as many OC projects have done over the last 2 decades. It was late in the project we realised that we could not fully explain why we were seeing intensity gains, and why the intensity gains largely did not correlate with any chemical shifts. This caused us to examine more carefully the effects of B1 inhomogeneity, and the conclusion presented that this compensation is responsible for the vast majority of our sensitivity gains. While from some perspectives this might be obvious, it took some time before this was obvious to us.

Finally, we anticipate that experimental people will play with these pulses in precisely the same way we did – via trial and error. While this is an universal experimental truth, we rarely see it acknowledged. So we prefer to leave this in.

Very little is written about how to choose number of offsets representing ppm ranges, number of elements in the Seedless pulse, its duration, width of interface regions, relative weights of simultaneously optimized criteria (regions with different aim) etc. Understanding the interplay among these parameters constitutes “the art of optimization”. Unexperienced user will never be optimizing pulses “on-the-fly”...

We respectfully disagree with the reviewer on this matter. We have now undergraduate students producing pulses on the fly in work with our lab.

To lower the barrier for entry, we have greatly expanded section S3 in the supplementary information. We specifically describe guidelines on setting the number of frequencies within a band, the duration of the finite element, the number of B1 fields when compensating for inhomogeneity, the effects of total duration / peak B1 field, how to monitor convergence and how to set up computations ab initio. We have included in the download

scripts that create pulses shown in the main text. We believe that this platform will let users to do this with limited training.

Further, provided that an experienced user has already made an input script for a pulse sequence, and provided the inexperienced user can calibrate the duration of 90° pulses on their sample, editing the values in the input script and then re-run Seedless, can certainly be done by an inexperienced user. We are presently in discussion with Bruker and Jeol about incorporation of our pulses into their operating systems, which in a best case means that re-calculating pulses could become invisible to a user and accomplished with a button push. From a strictly technical perspective, the barriers to take Seedless and implement in this way are modest.

Minor issues:

„In practical applications, a nucleus, peak field strength, amplitude (peak B1 field),...“ – repetition in the last two terms?

Fixed.

Seedless is sometimes typeset with small „s“ (seedless/Seedless)

Fixed. We have decided that Seedless should be a proper noun and therefore be capitalised always.

“...the need to account for undesirable uncontrolled evolution of chemical shift during pulses during delays ...“ – repetition

Fixed.

Why three levels of referencing in Figures? (Fig. 2 A ii)

We believe this layout makes the figure clearer. The wider views show the parts of the protein that are likely of experimental interest, but as described in the caption, we need to truncate the spectrum at the residual water signal is 2 orders of magnitude more intense than the protein resonances. The inset allows us to show precisely what is happening to the water.

Please read the Supplementary carefully again. There are many mistakes and typos...

We thank the reviewer for this comment and we now have refined the 85 pages of the supplementary section extensively. We note the breadth of the supplementary information, spanning detailed theory, guides for how to implement each pulse sequence, usage guides for Seedless, benchmarking against similar pulses and a complete synthesis for one of our ¹⁹F test molecules, and we hope this becomes a useful reference for anyone that wishes to understand the details.

Reviewer #2 (Remarks to the Author):

Andrew Baldwin and coworkers published a remarkable idea for optimal NMR spectroscopy.

It's claim that the parameter optimization is faster is not entirely clear to me and I have to trust. It is also unclear what the difference is to GRAPE's mathematical procedure. I did not find a discussion of convergence criteria. Pulses that are easy and fast to optimize but don't converge are potentially problematic in my experience. Particularly so if they are done on the fly. I generally trust that they tested the procedure.

I generally like that research and hope it lives up to its promises.

We thank the reviewer for their kind words. We too hope that this work lives up to what we have promised.

The reviewer is correct that there was no discussion on convergence criteria. We have included one now in section S3. In brief, we have never observed the phase only pulses to get stuck in a local minima on the way to the optimal solution (see response to reviewer 1). The software lets a user monitor the variation of infidelity with iteration number. This tends to plateau, indicating convergence has been reached. The value it converges on depends on the computation, where the more is asked of a pulse, the lower the final infidelity. In practical cases, we set a large maximum iteration value and wait to see what comes out at the end.

We hope that the download provided, complete with scripts that let users create the pulses we describe in the manuscript in a few seconds will let the NMR community bring this approach into their workflows.